



# Characterization and operation of a multi-channel Condensation Particle Counter (mc-CPC) for aircraft-based measurements

Sarah Richter[1], Timo Keber[1], Martin Heinritzi[1], Lisa Beck[1,2], Laurin Merkel[1], Sarah Kirchhoff[3], Jann Schrod[1], Patrick Weber[3], Joachim Curtius[1]

[1]Institute for Atmospheric and Environmental Sciences, Goethe University Frankfurt, Frankfurt am Main, 60438, Germany

[2]now at: Deutscher Wetterdienst, Offenbach am Main, 63067, Germany

[3]Institute of Climate and Energy Systems ICE-3, Forschungszentrum Jülich, Jülich, 52428, Germany

*Correspondence to*: Sarah Richter (richter@iau.uni-frankfurt.de)

**Abstract.** Field measurements of aerosol number concentration and aerosol size distribution in the upper troposphere and lowermost stratosphere (UTLS) are crucial for understanding the influence of processes such as new particle formation (NPF) on aerosol budgets, cloud formation and climate. In this study, we present the multi-channel Condensation Particle Counter (mc-CPC) that was designed and constructed for airborne measurements and tested during the TPEx campaign onboard a Learjet in 2024. The instrument uses FC-43 ($C_{12}F_{27}N$) as the working fluid and consists of three individual CPCs (Grimm SKY-CPC), a pressure regulation system and a common inlet. By varying the temperature difference ($\Delta T$) between each pair of saturator and condenser, the individual cutoff diameters ($d_{50}$) of each CPC can be adjusted. For the cases presented here, we typically operated two of the CPCs at a $\Delta T$ of 36°C for a direct comparison while the other CPC was set to a $\Delta T$ of 15°C. Two independent calibration setups were used to determine the cutoff and size-dependent counting efficiency of the mc-CPC at various internal and external CPC pressure levels. The experiments in the laboratory showed that the cutoffs of the individual channels were rather independent of the external pressure $p_{external}$ and only slightly dependent on the internal CPC pressure $p_{CPC}$, at least for a $p_{CPC}$ range between 200–350 hPa. A large fraction of flights during TPEx were conducted at an internal pressure of 250 hPa, and therefore the cutoff determined at 250 hPa was used as a fixed value for all internal pressures. For channel 1 and 2 that were operated at the same $\Delta T$, this gave a $d_{50}$ of 11.3 (±1.0) nm and 12.3 (±1.1) nm, respectively. Channel 3 was set to $\Delta T = 15$°C and a cutoff diameter of 14.9 (±1.3) nm was determined. In an internal pressure range between 200 hPa and 400 hPa the cutoffs decreased slightly with increasing $p_{CPC}$. Furthermore, our measurements also indicate that the cutoffs are not influenced by varying sample flows. The mc-CPC was operated for the first time on an aircraft during the TPEx campaign (TropoPause composition gradients and mixing Experiment) in June 2024. We present the first measurements of one research flight and discuss the uncertainties of the collected aerosol data.



## 1   Introduction

Aerosol particles play an important role in the atmosphere, e.g. for cloud formation and climate (Szopa et al., 2021). In the
planetary boundary layer (PBL) fine aerosols (PM$_{2.5}$) impact air quality and health (Cheng, 2014; Zhang et al., 2016;
Schraufnagel, 2020; Lee & Romero, 2023). In the free troposphere (FT) as well as in the upper troposphere (UT), aerosols
have an impact on the global radiation budget through the direct aerosol-radiation interaction and indirectly through aerosol-
cloud interaction (Peng et al., 2016; Lee & Romero, 2023). The latter describes the influence of aerosols mostly through their
role as cloud condensation nuclei (CCN) or ice nucleating particles (INP) on the formation and properties of clouds and their
impact on the radiation budget (Li et al., 2008; Wang et al., 2013; Seinfeld & Pandis, 2016). Aerosols can be directly emitted
(primary aerosols) or they can be formed from precursor gases as secondary aerosols (Seinfeld & Pandis, 2016). Freshly
formed nucleation mode particles (NMP) from new particle formation (NPF) are very small in size; here we define them in
the diameter range of 1–20 nanometers (Seinfeld & Pandis, 2016). A large fraction of the aerosols in the atmosphere originates
from the nucleation of vapors of ultralow or extremely low volatility (Donahue et al., 2011). To serve as CCN, these small
particles need to grow to sizes of at least 50 nm in diameter, which can happen through condensation of trace gases with
sufficiently low volatility and through coagulation (Merikanto et al., 2009; Seinfeld & Pandis, 2016; Gordon et al., 2017). It
has been estimated that approximately 50% of all CCNs in the troposphere result from nucleation (Merikanto et al., 2009; Yu
et al., 2014; Gordon et al., 2017). Hence, particle nucleation has a direct impact on the formation of clouds and therefore on
the climate of the earth, because microphysical cloud properties and the amount of cloud cover influence incoming solar
radiation and outgoing long-wave radiation as well as precipitation (Lee & Romero, 2023).

To gain a better understanding of the processes and components that drive particle nucleation and growth in the upper
troposphere and to reduce the uncertainties that these processes cause in current climate models, field measurements play a
significant role. Recent aircraft campaigns (Andreae et al., 2018; Williamson et al., 2019; Curtius et al., 2024) showed that
over tropical rain forest and the Pacific and Atlantic Ocean high numbers of several ten thousand aerosol particles per cm$^3$ are
frequently observed. These studies suggest that the high number concentrations are a result of new particle formation from
gas-phase precursors in the UT. Previous aircraft campaigns taking place in the northern hemisphere highlighted that Aitken
mode number particle concentrations in the middle and upper troposphere reach median values between 1000 and 1500 scm$^{-3}$, for N$_{>14\,nm}$ and N$_{>18\,nm}$ respectively (Schröder & Ström, 1996; Minikin et al., 2003) and in parts even > 10,000 scm$^{-3}$ (Hermann
et al., 2003). For nucleation mode particles the concentrations are even higher (Hermann & Wiedensohler, 2001; Minikin et
al., 2003; Rose et al., 2015). Various sources of NMP in the UT exist, but nucleation mode particles are dominated by local
production through new particle formation, defined as the combination of nucleation and initial growth. Upper tropospheric
NPF can occur under various atmospheric conditions in the mid-latitudes, for example in the outflow of convective systems
(Twohy et al., 2002), mixing of tropospheric and stratospheric air (Khosrawi & Konopka, 2003) or stratospheric air intrusion
(Zhang et al., 2024; Joppe et al., 2025). As observations of NMPs in the mid-latitude free troposphere and upper



troposphere/lower stratosphere (UTLS) regions are still sparse and the formation mechanisms are not well understood, it is crucial to extend these measurements onboard of aircraft.

To understand the mechanisms that drive NPF in the free troposphere and to estimate how large the initial growth is, aerosol
instrumentation is needed that reliably measures the aerosol number concentration as well as the size distribution under varying atmospheric conditions. Condensation particle counters (CPCs) are commonly used in aerosol science due to their ability to deliver reliable results of aerosol number concentrations at diameter sizes of a few nanometers at a fast response of $\geq 1$ Hz. To get a rough size distribution of aerosol particles at small sizes with d < 50 nm and a high time resolution several CPCs can be combined where each CPC becomes sensitive at a different particle diameter, the so-called cutoff diameters. The advantages
of this concept have been demonstrated in many recent studies using particle instruments for ground-based or aircraft-based measurements using several CPCs with different cutoffs, ranging from 3 nm to 60 nm (Dreiling & Jaenicke, 1988; Hermann & Wiedensohler, 2001; Minikin et al., 2003; Weigel et al., 2009; Rose et al., 2015; Williamson et al., 2018). Comparing the aerosol number concentration of different channels can provide valuable information about the location of nucleation events, the distribution and origin of NPF in the UT and the underlying aerosol growth processes.


Using a similar approach, we constructed a multi-channel Condensation Particle Counter (mc-CPC) for aircraft applications. The mc-CPC has three channels that are currently operated with FC-43 (Fluorinert) as the working fluid and which provide two different cutoffs by adjusting the internal CPC temperatures. A pressure regulation system with a critical orifice ensures a constant low pressure in the system. The instrument was used for the first time during the aircraft campaign TPEx (Tropopause
composition gradients and mixing Experiment). In this study, we describe the construction of the mc-CPC, present a detailed characterization of the cutoff diameters and show exemplary results from the Learjet TPEx campaign which took place in June 2024 in Hohn, Germany.

## 2    Methods

### 2.1    Description of the SKY-CPC and working fluids

For the construction of the mc-CPC we used three SKY-CPCs (Model 5411, Grimm Aerosol Technik GmbH, Germany). The SKY-CPC is a state-of-the-art condensation particle counter that has been designed for airborne applications. The measurement technique is based on the growth of aerosols through the condensation of a working fluid on an aerosol particle. The instrumentation consists of three modules. The saturator is held at temperatures $T_{sat}$ so that the working fluid (e.g. an alcohol) evaporates and gets mixed with the sample flow. In the condenser, the temperature $T_{con}$ is lower than $T_{sat}$ to create
supersaturation of the working fluid on the aerosol particles and thus enable their growth to optically detectable sizes. The third module is the photo-optical detection cell where the enlarged aerosols are detected through light scattering (Sinclair &





Hoopes, 1975; McMurry, 2000). Each particle that has a sufficiently large diameter is detected by the CPC. The smallest size at which particles are activated by the given working fluid is called the Kelvin equivalent size or critical diameter $d_{\text{kelvin}}$ (Seinfeld & Pandis, 2016). Aerosols that are smaller than this threshold value $d_{\text{Kelvin}}$ cannot be activated by the working fluid

due to the Kelvin effect. The aerosol diameter at which 50% of all aerosols are activated and measured is defined as the cutoff diameter or $d_{50}$. This diameter size depends on various conditions. For example the temperature difference between the saturator and condenser regime, the temperature dependent vapor pressure of the working fluid itself, the pressure in the system and the sample flow rate (Banse et al., 2001).

By default, the SKY-CPC is operated with 1-butanol (CAS: 71-36-3) as its working fluid and the saturator and condenser

temperature are set to 36°C and 10°C, respectively, which results in a cutoff diameter of 4 nm at ambient pressures of ~ 1000 hPa. We set two of the three SKY-CPCs to a saturator temperature of 41°C and a condenser temperature of 5°C, yielding a ΔT of 36°C. The third CPC was operated at 35°C and 20°C for saturator and condenser, respectively (ΔT = 15°C). The instruments have a constant flow rate $Q_{\text{CPC}}$ of 0.6 lpm that is maintained by a critical orifice, which is located downstream of the detection cell.

Butanol is a highly flammable and hazardous alcohol with a strong odor. Most of the CPCs used in ground-based research are running with butanol because it is well-proven for many applications as a reliable working fluid (Sem, 2002; Wlasits et al., 2020). It has a rather high vapor pressure of 6.7 hPa at 25°C (Roth, 2024), resulting in sufficiently low and well-defined cutoff diameters that increase at low pressures (Banse et al., 2001; Hermann et al., 2005; Weigel et al., 2009; Bauer et al., 2023). Due to its high flammability, aircraft certification of butanol for a CPC is challenging. Therefore, we decided to use

perfluorotributylamine (Fluorinert™ FC-43, 3M Performance Materials, St. Paul, Mn, USA, CAS: 311-89-7). Nevertheless, there are some disadvantages of FC-43 compared to butanol. For one, Fluorinert has an extremely high global warming potential (GWP) of ~7,000 (Hong et al., 2013), which demands responsible handling. The other disadvantage is the comparably low vapor pressure of 1.92 hPa at 25°C (3M, 2019). Even though the saturation vapor pressure at typical saturator temperatures for the two fluids are in the same order of magnitude (Table 1), the evaporation of FC-43 is lower, potentially suppressing the

activation of the aerosol particles. With a ΔT between saturator and condenser of 36°C, butanol can activate aerosols with a diameter of 2 nm whereas the particles that can be activated by FC-43 need to have a diameter of 5 nm (Hinds, 1999). However, some studies showed that the latter problem can be circumvented by decreasing the pressure in the CPCs, which facilitates the evaporation of FC-43, or by increasing the saturator temperature (Hermann et al., 2005; Gallar et al., 2006; Weigel et al., 2009; Williamson et al., 2018). FC-43 has already been used as a working fluid in several airborne CPCs. In Williamson et al. (2018)

cutoffs down to 3 nm could be realized by keeping the internal pressure constant at 120 hPa and by increasing ΔT to 36.4°C. Hermann et al. (2005) did a comparison between butanol and FC-43 operated TSI CPCs at different internal pressures, ranging from 200 to 1000 hPa. They demonstrated that the cutoff of the FC-43 CPC decreased with decreasing pressures while the maximum detection efficiency increased (highest counting efficiency compared to a reference instrument). The CPC operated with butanol showed the opposite effect in this study. Furthermore, Weigel et al. (2009) and Galler et al. (2006) also used

Fluorinert-operated CPCs at different pressures, but these were both custom made. Grimm SKY-CPCs have also been tested



at low pressures, but only with butanol, dimethylsulfoxide (DMSO) and water as a working fluid (Weber et al., 2023a; Weber et al., 2023b; Hermann & Wiedensohler, 2001; Bundke et al., 2015; Bauer et al., 2023). As far as we know, the application of FC-43 with a Grimm SKY-CPC has not been tested yet.

**Table 1: Characteristics of the working fluids Butanol and FC-43 and their performance on the CPC counting efficiency with regard to their saturation pressure. The vapor pressure at 36 and 10°C was calculated with the Antoine equation.**

| Parameter | Butanol | FC-43 |
|---|---|---|
| Chemical formula | $C_5H_{10}O$ | $C_{12}F_{27}N$ |
| Flash point (°C) | 35 | none |
| Boiling Point (°C) | 119 (@ 1 bar) | 174 (@ 1 bar) |
| Melting point (°C) | < -90 | -50 |
| Vapor pressure (hPa) | 6.7 (@ 25°C) | 1.92 (@ 25°C) |
| $p_{vap}$ (@ 36°C) (hPa) | 22.2 | 3.8 |
| $p_{vap}$ (@ 10°C) (hPa) | 4.2 | 0.7 |

## 2.2 Design of the multi-channel CPC

The multi-channel CPC (mc-CPC) was designed for the Learjet TPEx campaign. In Fig. 1 the flow schematic of the mc-CPC, including internal and external structure of the housing is depicted. The instrument consists of a 19-inch aircraft rack module including three individual SKY-CPCs (each 16.5 cm x 21.5 cm x 27 cm), accessory components, a bypass flow system to regulate and reduce the internal pressure and a common inlet system. The mc-CPC has a weight of 34.5 kg and the dimensions are 48 cm x 35 cm x 40 cm. The mc-CPC needs to be connected to an external pump to enable a constant flow through the

system. For the TPEx campaign, we used a dry scroll pump (IDP-3, Agilent IDP3D01). The mc-CPC as well as the pump are operated at a power supply voltage of 24 V.



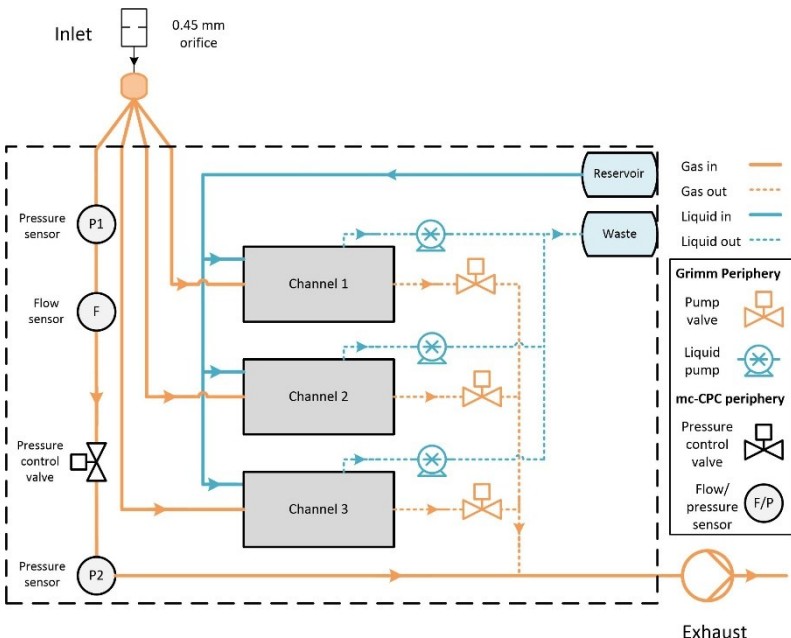

**Figure 1: Schematic of the mc-CPC containing three Grimm SKY-CPCs, a pressure regulation system and an inlet system with a**
**critical orifice. Blue lines represent tubes for liquid, and orange lines are for gases, respectively.**

Each SKY-CPC has a removable O-ring sealed 8 mm inlet tube that can be easily plugged in and out. To avoid leakages during
low pressures, we replaced the inlet with a leak tight fitting (1/4" NPT male to 1/4" tube). The SKY-CPCs were installed in
parallel into the rack. Because of the limited space in the housing, the length of each CPC inlet is different (Table A1).

The containers for the working fluid are made of PEEK material (Polyether ether ketone) and both, the reservoir and waste
container, have a volume of 200 ml and are attached to the front panel of the instrument (Fig. 1). They can be easily filled or
emptied via quick connectors. In order to provide an equal flow to the individual CPCs, we installed an inlet manifold, that
connects the individual CPC 1/4" sampling lines with the overall aerosol port, an 8 mm stainless steel tube. Due to space
restrictions, the sampling lines outside the mc-CPC housing needed to be strongly bent. Due to these circumstances, the total

lengths of the individual CPC inlet lines, measured from the inlet manifold to the CPC entrance were 58 cm (channel 1), 46.5
cm (channel 2) and 60 cm (channel 3) (Table B1). The data logging of the particle concentration and several other parameters
was realized using the Grimm nanoSoftware alongside a LabView-based custom solution.

For the pressure reduction we implemented an 8 mm ball valve in front of the mc-CPC common inlet, that included two
interchangeable orifices (0.45 and 0.65 mm). The bypass line of the pressure regulation system consists of two pressure sensors

P1 and P2 (Keller Model 23SY, see Fig. 1), a 10 lpm flow sensor F (Omron, Model D6F-10A6-000) and a 10 lpm solenoid
control valve (MKS, 248D). The pressure sensors are installed upstream and downstream of the MKS valve. The control valve
is regulated with a PID-controller on the pressure measured by P1, regulating the pressures $p_{CPC}$ inside the CPCs. P2 is
recording the pressure in the exhaust line. The flow sensor provides information about the additional flow that is needed to



ensure a low and constant pressure. The IDP-3 pump provides the flows $Q_{bypass}$ and $Q_{CPC}$ through the bypass and the SKY-
CPCs, respectively. For the pressure regulation and the data logging of the pressure and flow sensors we used a LabView
program.

### 2.3  Specifications of the TPEx campaign and the mc-CPC during research flights

#### 2.3.1 The TPEx aircraft campaign

The Tropopause composition gradients and mixing Experiment (TPEx) aircraft campaign took place at the airbase Hohn in
Germany. We conducted eight research flights and one test flight with the Learjet 35A in the time period from 03 June to 21
June 2024. One of the main goals of the campaign was the investigation of the extratropical UTLS region. In this context the
vertical transport of aerosols from the PBL into the UTLS as well as the effect of NPF was of special interest to us. For more
information regarding the TPEx campaign see Joppe *et al.* (2025); Breuninger *et al.* (2025); Bozem *et al.* (2025).


#### 2.3.2 Learjet 35A aerosol inlet and isokinetic sampling

For the aerosol instrumentation onboard the Learjet a dedicated aerosol inlet with a length of 620 mm was mounted through
an exchangeable window. It was designed and provided by the enviscope GmbH. The inlet tip is gold-plated and has a diameter
of 1.55 mm to ensure isokinetic sampling at a total flowrate of approximately 20 lpm and a true air speed (TAS) of 164 m/s.
Inside the Learjet a flowsplitter divides the sample flow into five aerosol inlets using four ¼" tubes placed around a 8 mm
aerosol inlet, which was used for the mc-CPC. The flows for all aerosol instruments and also the isokinetic flow regulation
system were provided by an IDP-3 dry scroll pump. The pump can maintain a maximum flow rate of 50 lpm over the entire
pressure range of the flight altitude. The isokinetic flow regulation system was operated with a 50 lpm MFC that maintained
an additional make-up flow according to the flight conditions.


#### 2.3.3 CPC inlet flow determination

During the TPEx reasearch flights the IDP-3 pump provided the flow for five aerosol instruments as well as for the isokinetic
flow regulation, which caused some difficulties for the critical flows through the CPCs. The required volume flow $Q_{CPC}$ of 0.6
lpm for each CPC could not always be maintained during the flights and was frequently lower. The inlet flow of the SKY-
CPC is controlled by a critical orifice that is located downstream of the detection cell. Unfortunately, the Grimm CPC does
not log the flow. As an internal verification of the sample flow rate the SKY-CPC only records the pressure drop ratio *pd* as
discrete values upstream and downstream of the orifice. For a choked or critical flow of 0.6 lpm, the pressure upstream of the





orifice needs to be about twice as high as the downstream pressure. If *pd* is below 2 however, the flow through the orifice is not critical anymore, resulting in erroneously calculated aerosol number concentrations, if no correction is applied. In order to

correct for this flow-related error we defined the Flow Factor *FF*, which is defined similar to *pd* but uses independent pressure measurements of the mc-CPC peripheral sensors to gain a more detailed understanding of the actual inlet flow:

$$FF = \frac{p_{upstream}}{p_{downstream}} = \frac{p_{CPC}}{p_{exhaust}}, \qquad (1)$$

The *FF* defines the relationship between the upstream and downstream pressure across the MKS control valve (P1 and P2 in Fig. 1), where the former represents the pressure in the measurement cell of a CPC ($p_{CPC}$) and the latter is the pressure in the exhaust line $p_{exhaust}$. The pressure $p_{CPC}$ is rather constant during measurement flights, changing only between 200 hPa and 350 hPa and is maintained by the mc-CPC bypass pressure regulation. However, the flow that provides the low pressures in the bypass (refers to *F* in Fig. 1) is not constant as it changes with the ambient pressure. The pressure $p_{exhaust}$ on the other hand is

dependent on the isokinetic flow in the Learjet inlet and thus on the TAS but also on $Q_{bypass}$. *FF* is therefore influenced by several variables and is subject to fluctuations. If *FF* is lower than 1.9, $Q_{CPC}$ is not critical anymore and thus smaller than 0.6 lpm which requires a correction of the particle concentration. We use the definition of a choked flow to calculate the corresponding correction factor $k_{FF}$ that was applied to all data when *FF* < 1.9. A comprehensive description of the flow correction is provided in Appendix B.


### 2.3.4 Data conversion and data flagging during TPEx

The conversion of the aerosol number concentration $N_i$ to standard temperature and pressure conditions STP was done using a two-step approach. This includes a first scaling of the mc-CPC pressure $p_{CPC}$ to the ambient pressure $p_{ambient}$ and a scaling of $T_{meas}$ (the temperature of the optics block of the CPC) to $T_{ambient}$. The ambient conditions were measured by the Learjet sensors.

The second step was the final STP conversion to $T = 273.15$ K and $p = 1000$ hPa (IUPAC). We also adjusted $N_i$ by the scattered light signal C1/C0 that is monitored in the CPC raw data. Here, *C0* and *C1* refer to a lower and higher detector threshold. This factor describes the behavior of particle growth in the CPC. A value of C1/C0 < 1 indicates that the growth and thus the diameter size of the aerosol particles is not sufficient to be counted. For a detailed description of this factor and the $N_i$ adjustment see Weber *et al.* (2023a) and Kirchhoff et al. (2025, in preparation). A particle loss correction in the inlet line or

the pressure-reducing orifice was not applied for most of the data points as we do not have detailed information about the aerosol size distribution that was present in our system during the research flights. Nevertheless, for selected periods with potential NPF events we implemented a particle loss estimation. Furthermore, a pressure-dependent correction of the counting efficiency of the three mc-CPC channels was applied (see 4.4).





After correcting the mc-CPC data we also applied a data flagging. A high fraction of 84% of all aerosol number concentrations measured during TPEx was influenced by the above-mentioned flow fluctuations (2.3.3). At values $FF < 1.2$ the data was dismissed due to high uncertainties in the correction factor $k_{FF}$; in this range a small change in $FF$ leads to large differences in $k_{FF}$. Data that were collected in the $FF$ range between 1.2 and 1.5 were flagged accordingly. The exact relation between $FF$ and the volumetric flow rate $Q_{CPC}$ can be found in Table B1.

Channel 1 and 2 of the mc-CPC were operated at the same saturator and condenser temperatures, which resulted in almost identical cutoff diameters (4.5). The two comparable channels were used as a measure to investigate the overall data qualitiy and consistency of the channels. For correct performance, the ratio Ch1/Ch2 should be close to 1. We assume that a non-systemantic point-by-point deviation of 20% is within the range of statistical uncertainties, which was calculated to a maximum 22% (see Appendix B). In the case of higher deviations, the data will be examined individually.


### 2.3.5 Identification of nucleation events

Another goal was to examine if new particle formation occurs in the extratropical UT similar to the tropics (Williamson et al., 2019; Curtius et al., 2024). The difference between the CPC channels 1 and 2 (lower cutoff) vs. channel 3 (higher cutoff) can give us an indication whether the air masses contained particles in the size-range between the two cutoff diameters, which 240 most likely have formed by recent NPF. Therefore, we used the following relation to define a new particle formation event (Weigel et al., 2009; Weigel et al., 2011; Curtius et al., 2024):

$$0.7 \cdot N_{small} - 1.3 \cdot N_{large} > 0 , \tag{2}$$

Here $N_{small}$ refers to the CPC channel with a smaller cutoff than $N_{large}$. To be identified as a potential NPF event, equation 2 has to be fulfilled for at least 10 seconds. By using this definition even systematic differences or a measurement uncertainty of 30% for each channel are not interpreted as NPF events.

### 3   CPC calibration

To characterize the three individual CPCs of the mc-CPC we used two different calibration setups, one being located at the 250 Goethe University Frankfurt (GUF) at the Institute for Atmospheric and Environmental Sciences (IAU). The second one was located at the Forschungszentrum Jülich (FZJ) at the Institute of Climate and Energy Systems (ICE-3), where we conducted the experiments at the IAGOS calibration lab. For both experiments we used the mc-CPC in its Learjet configuration with the 0.45 mm orifice upstream of the inlet lines. Both calibration setups consist of an aerosol generation system, an aerosol size



selection, the mc-CPC and a reference instrument. However, they differed in terms of the aerosol generation and composition,
the type of reference instrument and its internal pressure. Here, we provide a detailed description.

At both measurement sites, we characterized the mc-CPC at different internal pressures $p_{CPC}$ ranging from 200–750 hPa, where
the range of 200–350 hPa is of special interest for the aircraft settings. Nevertheless, we also did measurements at 500 hPa and
750 hPa to cover the whole range of free tropospheric conditions.

### 3.1 Calibration at GUF

#### 3.1.1 Calibration setup

In Figure 2 the schematic of the calibration setup for the mc-CPC characterization at GUF is given. The instrument was used
in the same setup as during the TPEx campaign (2.2).

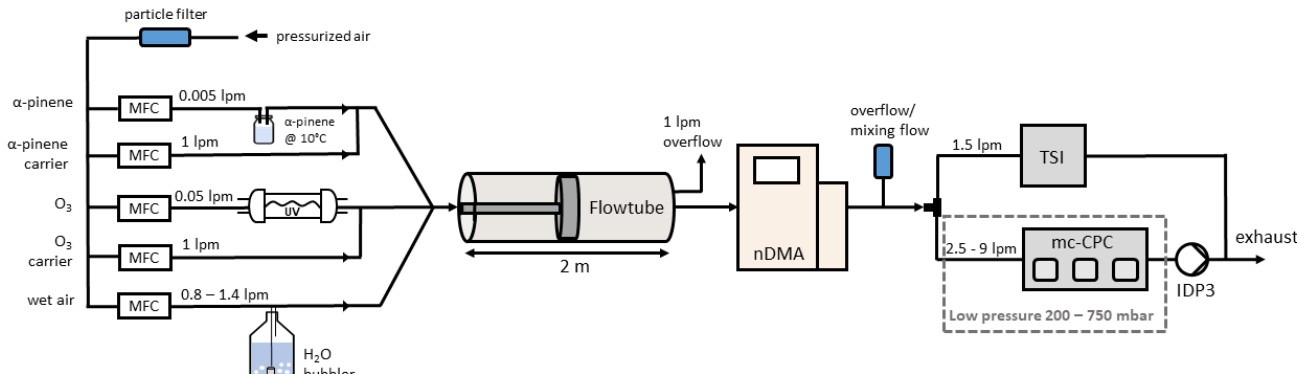

**Figure 2: Schematic of the calibration setup at GUF for the mc-CPC cutoff diameter determination, using α-pinene as an aerosol type. The mc-CPC was operated under low-pressure conditions (grey dotted line) while the TSI 3776 CPC measured at atmospheric pressure.**

The generation of the aerosol particles was realized via nucleation of α-pinene ozonolysis products by using a custom-made
flow tube. The flow tube setup consists of a 2 m stainless steel tube with a diameter of 72 mm, a set of five MFCs providing
different flow rates, a UV lamp for ozone generation and a $H_2O$ and α-pinene bubbler (Fig. 2).

For the setup, we used pressurized dry air that was filtered through a HEPA filter. The ozone was produced by flushing a UV
lamp at a flow rate of 0.05 lpm. A carrier flow of 1 lpm transported the ozone-air mixture into the chamber. For the α-pinene
(Sigma Aldrich, 98%) we used a cold reservoir at a constant temperature of 10°C. A small flow of 0.005 lpm was flushed over
the liquid and an additional carrier flow of 1 lpm introduced the mixture into the chamber. The wet flow was generated with a
0.8-1.4 lpm flow through the bubbler, which was filled with ultrapure water. For the latter setting we used an additional
overflow at the end of the flow tube to keep the sample flow constant. This adds up to a total flow of 3 lpm which was kept
constant for all experiments. With this setting a total particle number concentration up to 40,000 cm$^{-3}$ can be reached.





The flow tube is connected to an electrostatic classifier (TSI, model 3082) for a monodisperse sample. The aerosols are first charged in an x-ray neutralizer and are then size selected by their electrical mobility in the Differential Mobility Analyzer (nano-DMA, Vienna type). The recirculating sheath flow of the classifier was constant at 15 lpm for almost all measurements. This guaranteed a particle size selection of diameters between 3 nm and 60 nm. For an aerosol diameter of 80 nm as an upper limit, the sheath flow was reduced to 10 lpm, which is lower than the recommended ratio between sheath flow and aerosol flow of 1:5. Thus the error for this particle size was adjusted.

The monodisperse aerosol flow was divided between the mc-CPC and a reference CPC by a Y-splitter. We used a TSI model 3776 CPC as the reference instrument, which has a $d_{50}$ of 2.5 nm and an aerosol flow of 1.5 lpm. Together with the sample flow of the mc-CPC a maximum flow rate of 9.5 lpm was needed when operating the mc-CPC at 200 hPa. This results in a flow deficit of 6.5 lpm that was compensated by a particle free mixing flow implemented after the classifier. At higher internal pressures, the mc-CPC flow was correspondingly smaller.

As the sample flow of the TSI CPC is regulated by pressure differences along the sample path, it cannot be used in low-pressure regimes. It thus measured at laboratory ambient pressure throughout all experiments.

### 3.1.2 Evaluated parameters

The reference CPC was operated at ambient or external pressures of ~ 1000 hPa while the pressure inside the mc-CPC was adjusted based on the range of aircraft operation conditions during TPEx to 200 hPa, 250 hPa, 300 hPa and 350 hPa. Furthermore, we also did measurements at 500 hPa and 750 hPa to examine the whole range of free tropospheric pressures. The TSI CPC as well as the mc-CPC were STP corrected (IUPAC, 1983). Besides the internal pressure, also the inlet flow of the SKY-CPCs was adapted. At each internal pressure stage, an electrical mobility scan was performed at each sample flow listed in Table B1 and was then corrected with $k_{FF}$.

### 3.1.3 Measurement procedure

To reach a stable concentration and size distribution of α-pinene aerosol in the flow tube and in the particle counters, the flow tube was conditioned 45 minutes before the measurements. One measurement includes an aerosol size scan with the nDMA, having a step size of minimum 3 minutes and ranging from 3 nm to 80 nm. For each diameter step the first 15 and last 10 seconds were discarded to avoid a data distortion due to rapid and non-representative changes in the particle concentration. Each measurement cycle includes a background determination by turning off the voltages of the classifier.

We performed the experiment numerous times to cover a wide range of internal pressures and flows. The comparison of these two parameters was done in a pressure and flow range according to the TPEx flight conditions. Measurements at pressures higher than 350 hPa were only conducted for critical flows ($FF > 1.9$).

A particle loss correction calculated with the Particle Loss Calculator (PLC) (Weiden et al., 2009) was applied for the TSI reference CPC with regards to its inlet line to ensure a realistic counting efficiency. The mc-CPC setup for the Learjet was not



changed for the characterization in the lab. Here, we only calculated particle losses in terms of an additional inlet line that was installed to connect the mc-CPC with the reference instrument in order to characterize the conditions during the TPEx campaign. Because of the low-pressure regime in the mc-CPC system we adapted the PLC accordingly.

## 3.2 Calibration at FZJ

### 315 3.2.1 Calibration setup

The calibration setup at FZJ comprises an aerosol generation system, an nDMA for size selection, a mixing chamber and a Faraday Cup Electrometer (FCE) as well as a butanol SKY-CPC and the mc-CPC (see Fig. 3). For our experiments we used the FCE as a reference instrument.

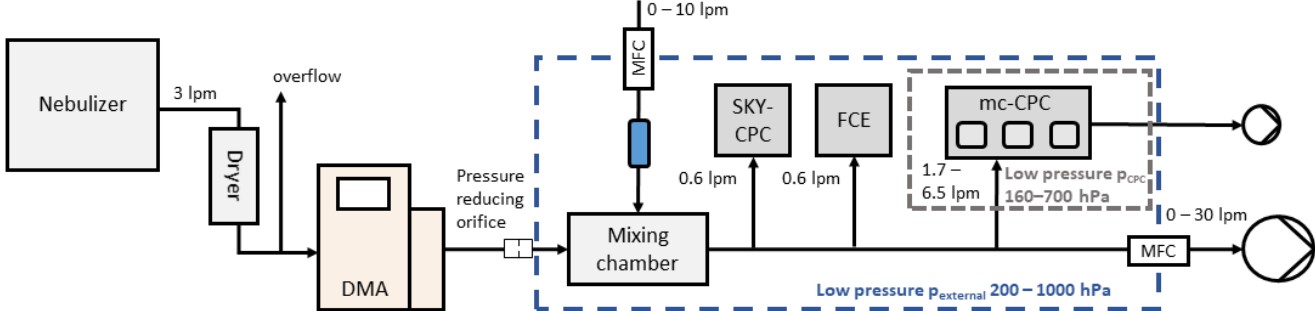


**Figure 3: Schematic of the calibration setup at FZJ for the mc-CPC cutoff diameter determination. Here NaCl was used as the calibration aerosol and a Faraday Cup Electrometer (FCE) as the reference instrument. The whole system was operated under low-pressures $p_{external}$ by using a critical orifice (blue dashed line). The pressure $p_{CPC}$ in the mc-CPC (grey dashed line) was adapted separately (adapted from Weber et al. (2023b)).**


We used NaCl as the calibration aerosol generated by a nebulizer with filtered air. The aerosol sample flow of 3 lpm was dried by a diffusion dryer. For particle selection we used a Vienna-type DMA (Model M-DMA 55-U, Grimm Aerosol Technik GmbH & Co. KG, Ainring, Germany) with a sheath flow of 6 lpm. With this, a lower and upper diameter size limit of 2.5 nm and 116 nm was achieved. A pressure-reducing orifice was installed downstream the DMA (Fig. 3), which leads to a pressure
reduction in the mixing chamber, where also the monitoring of the pressure took place. In the low-pressure mixing chamber, which has a volume of 500 ml, the monodisperse aerosol flow was mixed with a particle-free dilution flow. The latter ranged from 0-10 lpm and was adjusted by a MFC matching to the inlet flows of the particle instruments. The aerosol flow entered a sample line, from where it was distributed to the aerosol instruments. These consisted an FCE (Model 5.705, Grimm) and a butanol SKY-CPC (Model 5411, Grimm), of which we used the former one as the reference instrument, and the mc-CPC. All
instruments were able to measure at low pressures and had an inlet line of 25 cm. A comprehensive description of the calibration setup can be found in Weber et al. (2023b).



### 3.2.2 Evaluated parameters

At FZJ the sample flow $Q_{CPC}$ in the mc-CPC channels was constant at a targeted flow of 0.6 lpm throughout all experiments.

Similar to the measurements at GUF we changed the internal pressure $p_{CPC}$ according to the mc-CPC conditions during the research flights. This spans a pressure range of 200 hPa to 700 hPa for measurements including the pressure-reducing orifice. In addition to $p_{CPC}$ also the external pressure of the mc-CPC was adjusted to 300 hPa, 400 hPa, 550 hPa and 1000 hPa, which corresponds to the internal pressure of the FCE. These values were selected to mimic the ambient conditions during a research flight and to examine the influence of the pressure difference $\Delta p$ between $p_{external}$ and $p_{CPC}$ on the overall mc-CPC performance.

Another series of characterizations without including the orifice in front of the mc-CPC was carried out at 200 hPa, 300 hPa, 400 hPa, 550 hPa and 700 hPa, where $p_{CPC} = p_{external}$ to examine the particle losses due in the tube constriction and the pressure change.

### 3.2.3 Measurement and data analysis procedure

In the FZJ setup the particle source is primary particles which only need a short time to stabilize and to reach a constant signal in the aerosol instruments. The size selection of the DMA was carried out for 30s for each diameter or voltage level. Thereby the first 15s of each size selection were dismissed to avoid data impairment. A particle loss correction was applied to the 25 cm long inlet lines for the mc-CPC and the FCE data (Weiden et al., 2009). In case of the mc-CPC this includes particle losses for all given pressures $p_{CPC}$ and flow rates $Q_{CPC}$. The FCE was particle loss corrected according to the varying external pressures

$p_{external}$. In case of the FCE data, the following corrections were applied additionally: offset correction, multiple charge and flow rate correction  (Weber et al., 2023a). All mc-CPC data were STP corrected to $p_{external}$ and $T_{meas}$ (temperature in the optics block). We corrected the FCE data by the temperature of the laboratory $T_{lab}$. A pressure correction was not done for the FCE as it already operated at $p_{external}$.

### 3.3  Data evaluation methodology

To estimate the performance of a CPC, two parameters are commonly used. The counting or plateau efficiency $\eta_{max}$ and the cutoff diameter $d_{50}$. To derive the counting efficiency $\eta_{max}$ of the individual mc-CPC channel, the particle number concentration of the respective CPC at a specific diameter is compared to a reference instrument, in our case a TSI 3776 CPC (GUF) and an Grimm 5.705 FCE (FZJ). This leads to the following equation, which is dependent on the aerosol diameter $d_p$:




$$\eta_i(d_p) = \frac{N_i}{N_{TSI,FCE}} \ with \ i = channel \ 1, 2, 3 \tag{3}$$

The so-called cutoff curve, which can be retrieved from the comparison between the counting efficiency $\eta_i$ and the particle diameter, can be represented by a logistic sigmoid fit function:


$$\eta_i(d_p) = \eta_{max} + \frac{\eta_{min} - \eta_{max}}{1 + \left(\frac{d_p}{d_{50}}\right)^{d_0}} \tag{4}$$

Here $d_{50}$ refers to the aerosol diameter at which the counting efficiency between the mc-CPC channel i and the reference instrument reaches 50%. $d_0$ describes the onset diameter at which the particles get initially activated and $d_p$ is the aerosol diameter selected by the DMA. The efficiency $\eta_{max}$ stands for the plateau region, where the counting efficiency $\eta_i$ reaches its

maximum value and remains stable with increasing $d_p$. $\eta_{min}$ on the other hand refers to the lowest detection efficiency derived by equation 4 and is usually zero. Note that the sigmoid function can still be used to calculate the cutoff $d_{50}$ at 50% of the efficiency range between $\eta_{min}$ and $\eta_{max}$ even when $\eta_{max}$ is not 1, which is the case for many CPC calibrations (Hermann et al., 2005; Weigel et al., 2009). A discussion on this formula can be found in Appendix C.

## 4    Results and discussion

In this chapter we present several data sets of comprehensive measurements to characterize the mc-CPC with respect to the flight conditions as encountered during TPEx. The parameters we investigate are:

1) The influence of the CPC sample flow on the CPC performance
2) The influence of the internal CPC pressure on its performance for both calibration setups
3) The influence of the pressure reducing orifice on the CPC performance at various external pressures
4) The influence of internal and external pressure differences on the CPC performance

### 4.1  Influence of the sample flow

During the TPEx aircraft campaign, the flow rates of the SKY-CPCs were critical for only 16% of the measurements. This
raises two questions: 1) is the cutoff diameter influenced by the changing flows and 2) how is the data quality affected by it. To answer the first question we determined mc-CPC cutoff diameters at the GUF laboratory (section 3.1) for different combinations of four pressures and four flows.

 

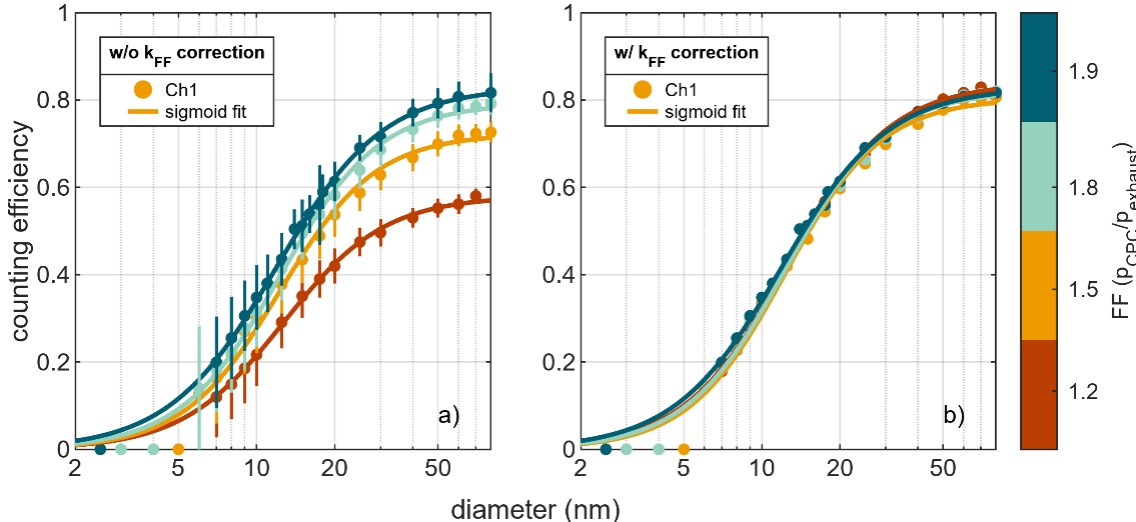

**Figure 4: Counting efficiencies of mc-CPC channel 1 with sigmoidal fit curves for $p_{CPC}$ = 250 hPa and $p_{external}$ = 1000 hPa and different flows (indicated as flow factor FF in the color bar). As a reference instrument a TSI 3776 was used which was operated at ambient pressure ($p_{external}$ = 1000 hPa). a) shows data without flow correction and respective fits, where b) is with flow correction and one fit curve including all data points measured at different sample flow rates. The error bars in a) refer to the averaged value and represent the standard deviation. Error bars were omitted in b) for sake of clarity.**

Figure 4 depicts the averaged counting efficiencies for each size bin from channel 1 at an internal CPC pressure of 250 hPa and an external pressure $p_{external}$ of 1000 hPa. In Fig. 4a the counting efficiency is displayed without the flow correction. The asymptotic counting efficiency $\eta_{max}$ for the lowest flow of 0.39 lpm or FF 1.2 reaches a maximum value of 58% only. The targeted and thus highest flow of roughly 0.6 lpm gives a maximum efficiency of 83% and a $d_{50}$ of 11.9 nm. Figure 4b represents the same data as Fig. 4a but includes the flow correction (as described in Appendix B). All data points that were measured at FF < 1.9 are now in line with the target cutoff curve at FF = 1.9 (dark blue). Here, the difference in the asymptotic efficiencies is less than 5% between the individual cutoff curves. Regarding the cutoff diameter, the largest difference of 0.6 nm occurs between the target flow and FF = 1.2, where the cutoff is slightly higher for lower flows. Still, the deviation is small and within the uncertainty, pointing to the importance of applying the flow correction. This result is also representative for channel 2 and 3 of the mc-CPC. This leads to the conclusion that the prolonged residence time of the aerosols in the CPC system is not affecting the cutoff diameter. Furthermore, the results demonstrate that the flow correction is valid for the laboratory calibration measurements and that the reduction of the sample flow has a negligible influence on the $d_{50}$ and was therefore disregarded. Consequently, the cutoff characterization in the following plots is only discussed for the flow corrected parameters.

**4.2 Comparison of the two calibration setups**

Here, we present the results of the mc-CPC characterization at the Goethe University Frankfurt (GUF) and at the Forschungszentrum Jülich (FZJ). In Fig. 5a the cutoff measurements of channel 1 at GUF are shown at a constant external pressure of 1000 hPa and a varying $p_{CPC}$ in the range of 200 hPa to 750 hPa. Figure 5b depicts comparable measurements from



FZJ with a $p_{external}$ of 400 hPa and $p_{CPC}$ from 200 hPa to 315 hPa. All data points shown in Fig. 5 and also in the following plots
are normalized by their plateau efficiency $\eta_{max}$. The cutoff diameters and plateau efficiencies derived from the raw data are
listed in Table 2. The focus here is on the pressure range between 200 hPa and 350 hPa, as these were the pressures at which
the mc-CPC was operated during the TPEx campaign. To examine the performance of the instrument at lower to middle
tropospheric conditions, we included also pressures of 500 hPa and 750 hPa at GUF. Note that the differences in the
measurement setups arise due to technical limitations, which mainly concern the differences in the external pressures and the
test aerosol.

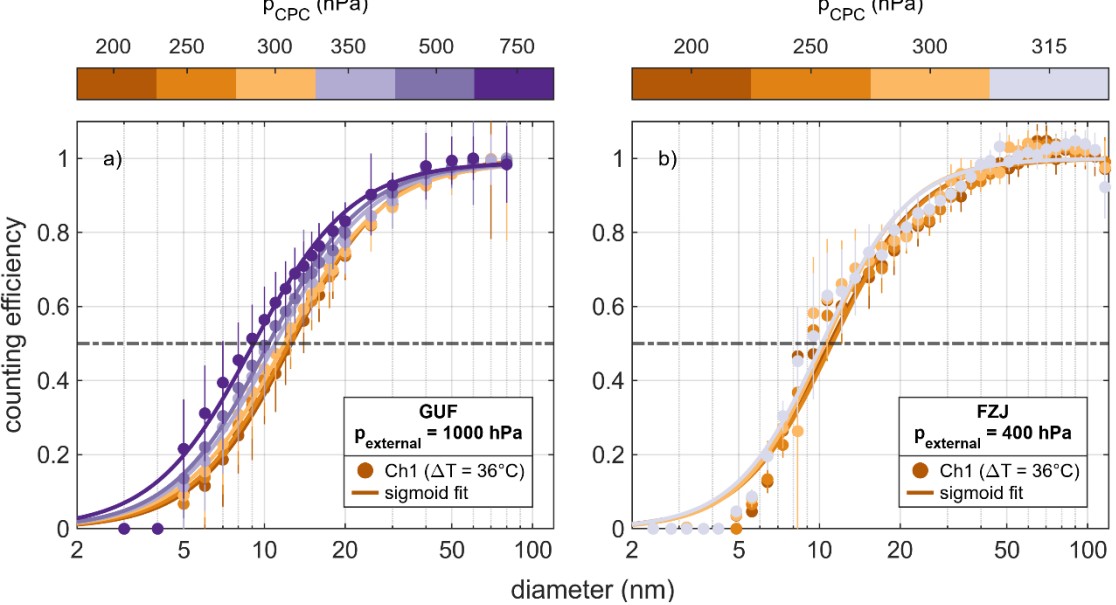

**Figure 5: Counting efficiencies normalized by the plateau efficiency for channel 1 of the mc-CPC measured a) at GUF with a constant**
**external pressure of 1000 hPa and a TSI 3776 reference instrument, where measurements at different sample flows are averaged**
**and b) at FZJ with a constant $p_{external}$ = 400 hPa using an FCE as reference instrument. Both with varying internal pressures (as**
**indicated by the color bar). The fits are derived from Eq 4. The error bars are the standard deviation of each CPC diameter step.**

The direct comparison between the two calibration setups of Fig. 5a and 5b shows that the overall fit progression follows a
similar trend. Still slight differences in the plateau efficiencies as well as in the cutoffs are visible. However, the measurements
taken at FZJ show much higher efficiencies throughout all pressure stages compared to the GUF measurements, i.e. when the
data is not normalized by $\eta_{max}$ (see Table 2). At a $p_{CPC}$ of 250 hPa $\eta_{max}$ is 94% for FZJ and 82% for GUF, respectively. This
difference of roughly 10% stays rather constant at comparable internal pressures, which results in a difference in the cutoff
diameter as well. When obtaining the cutoff diameters from Figure 5a for the GUF measurements at 250 hPa internal pressure,
channel 1, 2 (operated at $\Delta T = 36°C$) and 3 (operated at $\Delta T = 15°C$) reached cutoff diameters of 12.4 nm, 12.1 nm and 16.4
nm, respectively (Figures for channels 2 and 3 are presented in Appendix D). A $d_{50}$ error describes the bandwidth of the DMA
transfer function. The experiments at FZJ for the same channels showed $d_{50}$ values of 11.2 nm, 12 nm and 14.7 nm. The cutoff



diameter is therefore in agreement for both setups for all three channels when considering the uncertainties $\Delta d_{50}$ of the cutoff determination listed in Table 2.

**Table 2: Cutoff diameters ($d_{50}$) and plateau counting efficiency $\eta_{max}$ derived from Eq. 4 for channel 1 ($\Delta T = 36°C$) measured with the calibration setup at Goethe University Frankfurt (GUF) and Forschungszentrum Jülich (FZJ). The parameters determined for $p_{CPC} = 250$ hPa at FZJ are the average of two measurements and the GUF data are averaged values over all measurements at different flows. The $d_{50}$ error is given as the DMA mobility bandwidth error.**

| | GUF ($p_{external} = 1000$ hPa) | | FZJ ($p_{external} = 400$ hPa) | |
| --- | --- | --- | --- | --- |
| $P_{CPC}$ (hPa) | $d_{50}$ (nm) | $\eta_{max}$ (%) | $d_{50}$ (nm) | $\eta_{max}$ (%) |
| 200 | 12.8 (±1.3) | 80 | 10.9 (±0.9) | 90 |
| 250 | 12.4 (±1.3) | 82 | 11.2 (±1.0) | 94 |
| 300 | 11.7 (±1.2) | 84 | 10.5 (±0.9) | 95 |
| 315 | | | 10.2 (±0.9) | 96 |
| 350 | 10.9 (±1.1) | 85 | | |
| 500 | 10.3 (±1.1) | 84 | | |
| 750 | 9.1 (±1) | 94 | | |


Both laboratory experiments show that with increasing internal pressures the plateau efficiency increases while the cutoff decreases, but this effect is stronger at GUF than at FZJ (Table 2). This is true for all three channels in the given pressure range of 200-350 hPa. For the FZJ experiments the cutoff diameter shifts by ~1 nm to smaller values when increasing the pressure from 200 hPa to 315 hPa, where the differences are in the range of uncertainty. At GUF we saw a decrease of less than 2 nm

in $d_{50}$ while changing $p_{CPC}$ from 200 hPa to 350 hPa, though the differences become larger when increasing the pressure even further. This is similar to the results from Bauer *et al.* (2023) who noted the same trend with identical Grimm instruments using butanol as the working fluid. Previous studies using FC-43 with CPCs from other manufacturers indicate that the cutoff of the respective instrument shifts to smaller sizes at lower internal pressures (Hermann et al., 2005; Weigel et al., 2009; Williamson et al., 2018).

Figure 5 shows that the increase of the CPC pressure leads to an enhanced CPC performance and this behavior was reproduced for both experimental setups. Furthermore, the $d_{50}$ changes only slightly for the $p_{CPC}$ pressure range of 200–350 hPa. At FZJ, we also had the opportunity to characterize the mc-CPC regarding the flight conditions during TPEx. Here we also changed the external CPC pressure to mimic different flight levels, which could not be done at GUF. As the measurements carried out at FZJ are thus more representative for the conditions during the TPEx flight campaign, we will proceed with discussing these

results, but comparable measurements, which were performed at GUF, are documented in the SI for completeness.

We will use cutoff diameters obtained at $p_{CPC} = 250$ hPa for all reasearch flights, as this was the most commonly selected internal pressure (46% of all in-flight data). Although the cutoff diameters are changing by ~1 nm with different $p_{CPC}$ the ratio



between the channels with smaller cutoff diameters (1 and 2) and the channel with larger cutoff (3) stays rather constant within
1 nm (Table D2).


## 4.3 Tests without pressure reduction by critical orifice

One reason for the discrepancy between the data shown in Fig. 5 and former studies could be caused by the particle losses in
the inlet system due to the pressure-reducing orifice. To verify how large its influence on the cutoff diameter and the counting
efficiency of the CPCs is, we removed the orifice and did diameter scans for 200 hPa, 300 hPa, 400 hPa, 550 hPa and 700 hPa,

which represents flight altitudes between 11,500 m and 3,000 m. For these measurements the electrometer and the mc-CPC
were operating at the same pressures, where $\Delta p = p_{external} - p_{CPC} = 0$.

**Table 3: Cutoff diameters and counting efficiencies of mc-CPC channel 1, 2 and 3, operated without pressure-reducing orifice for
varying pressures. FCE is used as the reference instrument. The values listed are derived from Eq. 4. The $d_{50}$ error is given as the**
**DMA mobility bandwidth error.**

| $P_{external} = p_{CPC}$ (hPa) | Channel 1 ($\Delta T = 36°C$) | | Channel 2 ($\Delta T = 36°C$) | | Channel 3 ($\Delta T = 15°C$) | |
|---|---|---|---|---|---|---|
| | $d_{50}$ (nm) | $\eta_{max}$ (%) | $d_{50}$ (nm) | $\eta_{max}$ (%) | $d_{50}$ (nm) | $\eta_{max}$ (%) |
| 200 | 11.9 (±1.0) | 95 | 11.7 (±1.0) | 97 | 13.5 (±1.2) | 101 |
| 300 | 10.2 (±0.9) | 100 | 9.4 (±0.8) | 102 | 12.7 (±1.1) | 105 |
| 400 | 9.9 (±0.9) | 99 | 10.0 (±0.9) | 100 | 12.7 (±1.1) | 100 |
| 550 | 8.2 (±0.7) | 99 | 9.0 (±0.8) | 98 | 22.2 (±2.0) | 101 |
| 700 | 8.2 (±0.7) | 99 | 8.8 (±0.8) | 100 | 39.0 (±3.5) | 56 |



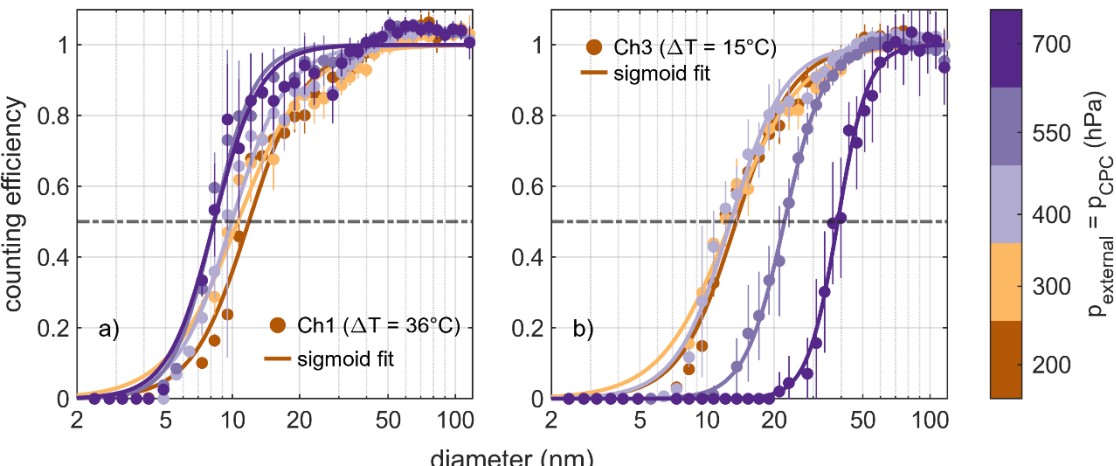


**Figure 6: Normalized counting efficiency for a) channel 1 and b) channel 3 derived from FZJ measurements at different ambient pressures (colorbar) with FCE data. The pressure-reducing orifice was not used during these experiments, therefore internal and external pressure are the same. The error bars are the standard deviation of each CPC diameter step.**

In Fig. 6a and 6b the normalized counting efficiencies for channel 1 ($\Delta T = 36°C$) and channel 3 ($\Delta T = 15°C$) are depicted, respectively, color-coded by the pressure. As channel 2 has the same behavior as channel 1, it was not included in Fig. 6 but the results are included in Table 3. Similar to Fig. 5b, the dependence of the pressure on the CPC performance changes, at least for channel 1 and 2. For 200 hPa we determined a $d_{50}$ of 11.9 nm, 11.7 nm and 13.5 nm for all three channels, which leaves only a difference of less than 3 nm between the 'small' and the 'large' CPC channels. Overall, the decrease of the cutoff

diameters in the relevant pressure range of 200 hPa to 400 hPa is smaller than 2 nm for channel 2 and 3, which is comparable to the results from the measurements with the orifice installed (Table 2). The smallest cutoff diameters of 8.2 nm and 8.8 nm were reached for channel 1 and 2 at 700 hPa. During the pressure increase, the counting efficiency for the two channels only changes slightly by 5%. However, this behavior is not fully reproducible for channel 3. At 550 hPa and 700 hPa, the trend reverses and the onset diameter as well as the cutoff moves to larger diameters. From the raw data in Table 3 we can also

determine that the plateau efficiency stays rather constant between 200 hPa and 550 hPa but decreases drastically to 56% at 700 hPa. We cautiously suggest that the drop in the CPC performance at 700 hPa is due to a reduced FC-43 diffusion rate in the saturator and a small $\Delta T$. The diffusion rate is highly dependent on the pressure, being enhanced at lower pressure levels. By increasing $p_{CPC}$ to 700 hPa we possibly also decreased the diffusion of FC-43 into the center of the saturator. This could have led to areas with only small or no supersaturation, which consequently could have had an unfavorable effect on the

activation of the particles. The much higher saturator temperature of channel 1 and 2 may have balanced out the effect of the high pressure. Note, that these are assumptions, which need further investigation.

The cutoff diameters determined for channel 1 and 2 without the orifice are very similar to the ones for the default mc-CPC setup. The counting efficiency on the other hand is higher when measuring without the orifice. Table 3 shows that for all

channels the maximum efficiency of 100 ($\pm$5) % is reached for almost all pressure stages. In Fig. D1-D3 (or Table 3) $\eta_{max} = 1$





is not reached. This suggests that the critical orifice installed before the mc-CPC inlet causes particle losses. Especially particles in the size range were the plateau efficiency reaches its maximum seem to be affected. This could explain the small difference between the cutoff diameters measured with and without orifice and the larger differences regarding the plateau efficiency. Nevertheless, the trend of an increased CPC performance at increased pressures is reproducible even without the orifice, which leads to the conclusion that the limiting factor is not the particle loss through the orifice but the geometry of the inlet line.

### 4.4 Influence of the external CPC pressure

In Fig. 7 the normalized counting efficiencies of all three channels at a constant internal CPC pressure of 250 hPa and at four external pressures are depicted. The cutoff diameters as well as the plateau counting efficiency of the raw data are listed in Table 4. For the graphical representation of the raw mc-CPC data, see Appendix E.

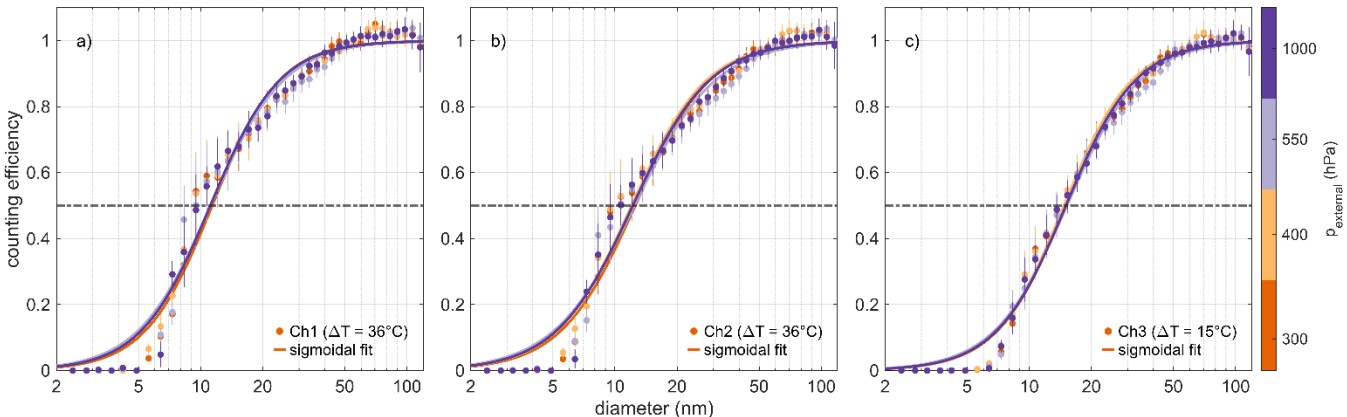

**Figure 7: Counting efficiencies normalized by the plateau efficiency of all three mc-CPC channels (a – c), color-coded by the external pressures. The internal pressure $p_{CPC}$ is 250 hPa for all panels. The error bars represent the standard deviation. Note that for $p_{external}$ = 300 and 400 hPa the depicted data is averaged for two measurement cycles.**

From Table 4 one can directly derive a dependency of the counting efficiency with the external pressure, which is reproducible for all three mc-CPC channels. The higher $p_{external}$ and thus the higher $\Delta p$, the lower is the plateau efficiency. This can probably be explained by increased inlet line losses resulting from the higher pressure drop and thus the higher flow that is provided by the mc-CPC bypass system. However, the differences between pressure levels are rather small. The smallest counting efficiency could be derived at $p_{external}$ 1000 hPa being the same for each channel at 87%. The maximum $\eta_{max}$ is associated with





the lowest $p_{external}$ of 300 hPa ($\Delta p = 50$ hPa) and gives values from 94 to 96% for channel 1–3, respectively. In total the change in efficiency is less than 10% for all channels in the external pressure range of 300 hPa to 1000 hPa.

Note that the cutoff diameters determined for the different pressures are relatively constant. In case of channel 1, $d_{50}$ ranges from 11.1 nm to 11.4 nm which is within the uncertainty, while the deviations for channel 2 were somewhat larger, ranging from 12 nm to 12.6 nm, but without having a clear increasing or decreasing trend with $p_{external}$. Channel 3 on the other hand is

changing from 14.7 nm to 15.2 nm, which is again within its uncertainty. The differences in the channels are probably due to statistical deviations.

**Table 4: Cutoff diameters and counting efficiencies (Eq. 4) of mc-CPC channel 1, 2 and 3, operated in the Learjet configuration for varying external pressures and at a fixed internal pressure of 250 hPa. The FCE was used as the reference instrument. The $d_{50}$ error**
**is given as the DMA mobility bandwidth error.**

| | Channel 1 ($\Delta T = 36°C$) | | Channel 2 ($\Delta T = 36°C$) | | Channel 3 ($\Delta T = 36°C$) | |
|---|---|---|---|---|---|---|
| $p_{external}$ *(hPa)* | $d_{50}$ *(nm)* | $\eta_{max}$ *(%)* | $d_{50}$ *(nm)* | $\eta_{max}$ *(%)* | $d_{50}$ *(nm)* | $\eta_{max}$ *(%)* |
| 300 | 11.4 (±1.0) | 94 | 12.6 (±1.1) | 94 | 15 (±1.3) | 96 |
| 400 | 11.2 (±1.0) | 92 | 12.0 (±1.0) | 92 | 14.7 (±1.3) | 93 |
| 550 | 11.2 (±1.0) | 90 | 12.4 (±1.1) | 89 | 15.2 (±1.3) | 91 |
| 1000 | 11.1 (±1.0) | 87 | 12.2 (±1.1) | 87 | 15 (±1.3) | 87 |

The measurements indicate that the cutoff diameters are not strongly dependent on the external pressures and also the plateau efficiency is only to a minor degree influenced by $p_{external}$. In the case of aircraft measurements, this means that varying altitudes

do not alter the cutoff but the plateau efficiency changes slightly with altitude and therefore needs to be corrected for accordingly.

### 4.5 Synthesis of the laboratory measurements

Figure 8a depicts the cutoff diameters of the three CPCs at varying internal pressures. The color code refers to the $\Delta p$ between upstream ($p_{external}$) and downstream ($p_{CPC}$) pressure of the critical orifice. With an increase in CPC pressure the cutoff decreases.

In the displayed measurement range of 160 hPa to 700 hPa the cutoffs of channels 1 and 2 decrease by about 5 nm. This relation arises for channel 3 as well, but only at lower pressures of up to 400 hPa. The cutoffs of all channels seem to be rather independent of the pressure change between $p_{external}$ and $p_{CPC}$ although the counting efficiency shows higher values when $\Delta p$ is low. During the TPEx campaign the pressure difference between the ambient and the CPC pressure was most of the time lower than 200 hPa which leads to a difference in $\eta_{max}$ of 5–10% (Fig. F1). An aerosol number concentration correction

regarding varying external (or ambient) pressures was done for the measurements during the research flight presented in chapter 5.




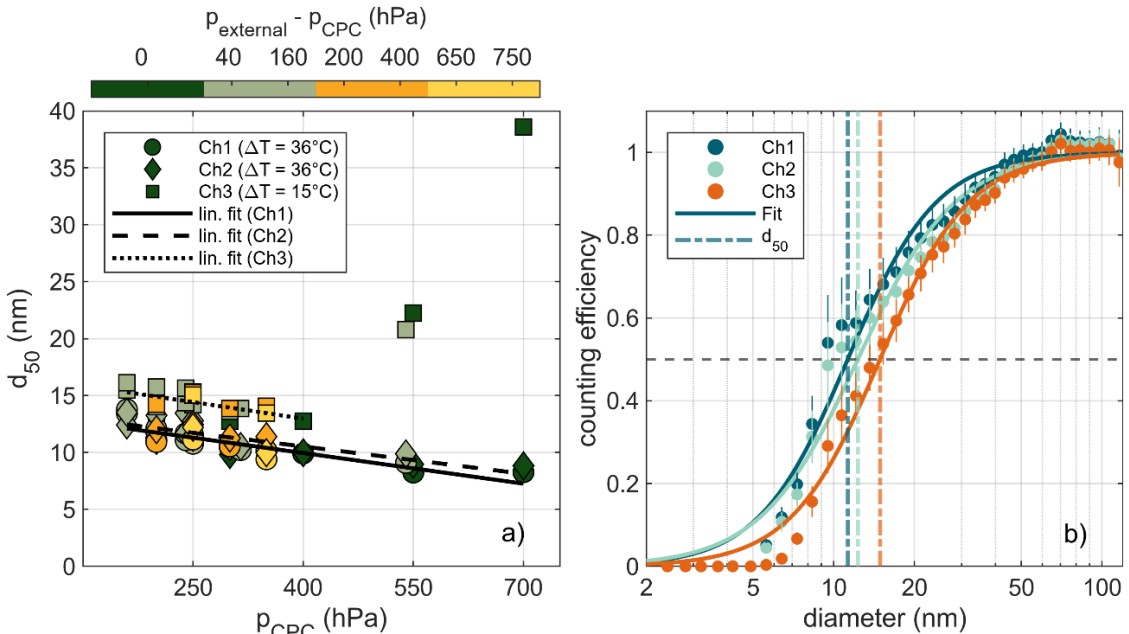

**Figure 8: a) FZJ measurements for all mc-CPC channels plotted as cutoff diameters dependent on $p_{CPC}$, color coded by $\Delta p = p_{external} - p_{CPC}$; and b) normalized counting efficiencies with corresponding cutoff diameters measured and averaged at $p_{CPC}$ = 250 hPa and $p_{external}$ = 300 and 400 hPa. Error bars indicate the standard deviation.**

Looking at the internal pressure range of 200–350 hPa it becomes apparent that the cutoff changes only slightly. For Channel 1 this means a decrease of ~1 nm with increasing $p_{CPC}$, channel 2 drops about 1.6 nm and channel 3 decreases 1.2 nm. This indicates that the overall decrease in cutoff with $p_{CPC}$ is rather constant for all channels, at least in the relevant pressure range. We therefore decided to use only one characteristic cutoff diameter per channel for the pressure range of 200 – 350 hPa. With the most commonly observed pressures of $p_{CPC}$ at around 250 hPa and $p_{external}$ of 300–400 hPa (23%), the cutoffs are determined to be 11.3 nm regarding channel 1, 12.3 nm for channel 2 and 14.9 nm for channel 3, as shown in Fig. 8b. These values represent mean values of two measurement sets for $p_{external}$ of 300 hPa and 400 hPa in each case at two different days. The errors are the standard deviation of these four combined measurements. The channels 1 and 2 have the same $\Delta T$ of 36°C between their saturator and condenser temperature, which should in theory result in a comparable cutoff. With our measurements, we show that this is true within the range of uncertainty. Note that the cutoff difference between the smallest and the largest channel is only 3.6 nm. This is rather small regarding the large difference in $\Delta T$. We also observed that the performance of the CPC (cutoff diameter and plateau efficiency) is decreasing at low pressures and not the other way around, as Hermann *et al.* (2005), Weigel *et al.* (2009) and Williamson *et al.* (2018) presented. As another striking feature it needs to be pointed out that at a pressure of 550 hPa the cutoff of channel 3 suddenly increased with the pressure, which we have not seen for the other channels. First, we want to emphasize that the previous studies that investigated the enhanced efficiency of FC-43 at lower pressures used different CPC instruments than we did (e.g. TSI CPCs). Furthermore, there are studies that



examined the behavior of Grimm CPCs at low pressures (Weber et al., 2023b; Bauer et al., 2023) and observed a similar relationship as we did, but were using butanol as the working fluid. Due to the different measurement setups, these studies cannot directly be compared to ours. Currently, we can only speculate about the reasons for this behavior in our system. We tentatively propose that altering diffusion rates in combination with the long mc-CPC inlet lines could have caused the dropping CPC performance with decreasing pressures. The diffusion of gases is dependent on the temperature and the pressure; low

pressures and high temperatures are most favorable. On the other hand, the diffusion losses of aerosols to the wall are also affected by these parameters and additionally by the length of the inlet lines. The diffusion losses of small particles are high at low pressures (in the size range of 2–35 nm five times higher for 200 hPa than for 1000 hPa). Our inlet line is rather long, which enhances the particle losses even further. This could explain the large cutoffs that we observed even at the highest ΔT. When increasing the pressure in the range of 200 hPa to 400 hPa, the aerosol diffusion coefficient decreases, which could have

led to lower cutoff diameters. We assume that saturator temperatures of 41°C and 35°C, respectively, in this pressure regime are high enough for the FC-43 to reach the center of the saturator and to reach supersaturation (Hermann et al., 2005). A further increase of $p_{CPC}$ could have caused unsaturated sections in the saturator due to a lowered FC-43 diffusion rate (Hermann et al., 2005; Bauer et al., 2023). This seems to have an effect only on the performance of channel 3, as the cutoff diameters of channel 1 and 2 still decrease with increasing $p_{CPC}$ (Fig. 8a).


       With these results, we can conclude that the mc-CPC in this configuration is best suitable for low internal pressures. For aircraft campaigns that focus on different flight levels, a pressure regulation is suitable. Especially when flying in the UTLS region it might be appropriate to remove the orifice and thus the pressure regulation to avoid the additional particle losses.

## 5    First results of TPEx

During the TPEx campaign we performed eight research flights (RF) in different regions of Germany, the North Sea and the Baltic Sea. Figure 9 shows the time series of several variables measured during RF04 on 12 June 2024. The take-off and landing took place at the airbase in Hohn, Germany. The flight was 3.5 hours long and was conducted over the Baltic Sea in a northerly direction towards Sweden. In the following section, we will discuss the factors that could influence the particle number concentration and related variables measured by the mc-CPC. An in-depth discussion of the aerosol data and possible

atmospheric implications is beyond the scope of this study and will be presented elsewhere.



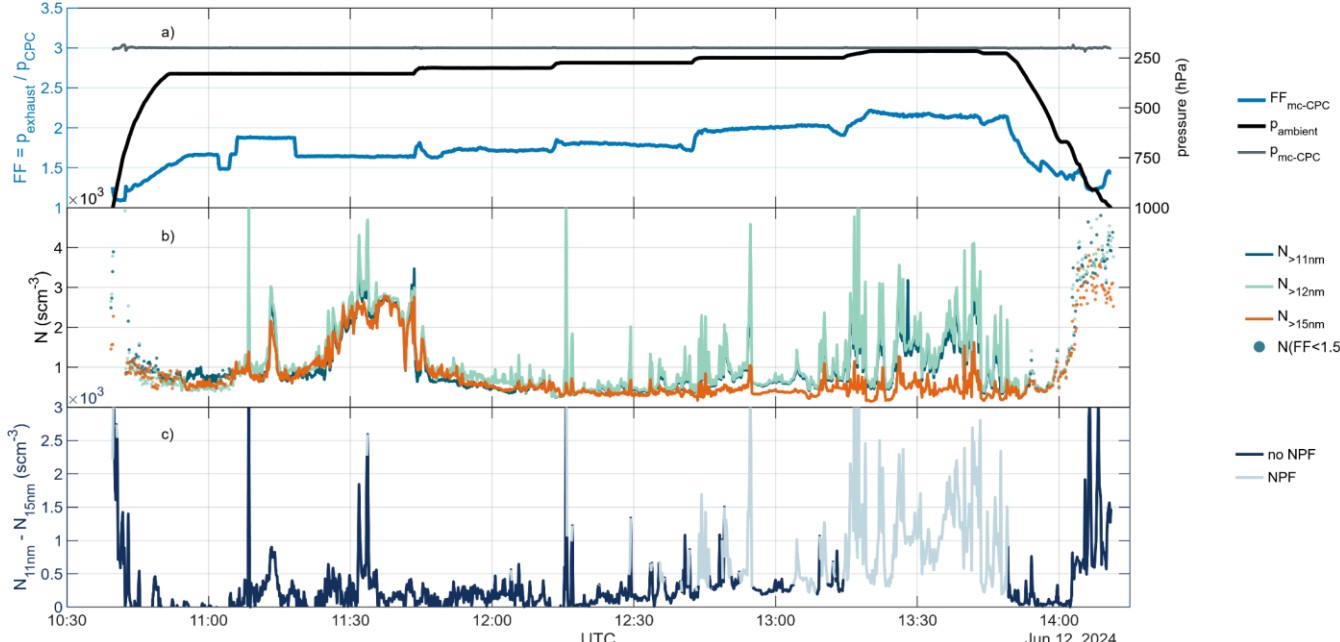

**Figure 9: 10 s average time series of different parameters during the research flight 4 (RF04) on 12 June 2024 starting from Hohn, Germany with the Learjet 35A. a) internal pressure of the mc-CPC system, the ambient pressure and the Flow Factor (FF = $p_{exhaust}$/$p_{CPC}$) which transfers to the sample flow, b) aerosol number concentration (at STP) for three CPC channels ΔT(channel 1&2) = 36°C, ΔT(channel 3) = 15°C) and flagged data (FF between 1.2-1.5) is represented by dots and c) aerosol number concentration between channel 1 and 3, colored in light blue are areas with potential NPF events (Equation 2).**

The internal pressure $p_{CPC}$ measured by the mc-CPC peripheral system was set to 200 hPa during RF04. Fig. 9a shows that this value could be maintained during the whole flight, even in the altitude transition regimes. Note that the flight altitude varied only between 330 hPa and 215 hPa, which is favorable for a constant $p_{CPC}$ of 200 hPa. The flow factor $FF$ in Fig. 9a represents the sample flow rate of the CPCs. In the time between 10:40 and 12:30 UTC the $FF$ value is most of the time below a value of 1.7 (refers to a flow of < 0.55 lpm). This indicates that the flow through the CPCs during this time was neither critical nor constant. $FF$ is strongly dependent on the flight altitude and pump performance, but it steadily increases with flight altitude, approaching a critical constant flow. Between 12:35 and 13:55 UTC $FF$ reached a value > 1.9 which leads to a critical flow of 0.6 lpm through the individual CPCs.

The particle number concentration of the three channels in Fig. 9b shows close agreement over most of the first half of the flight. In Fig. 9c, the difference in particle concentration $N_{11-15}$ between channel 1 and 3 underlines the similarity between the channels. During the first half of the flight $N_{11-15}$ is often below 500 scm$^{-3}$ and rather constant, showing only a few higher concentration peaks. In addition, the NPF criteria Eq. 2 (light blue markers in Fig. 9c) was only fulfilled for a few seconds, which indicates that in this part of the flight the aerosol particles were mainly Aitken mode particles and not freshly formed. However, in the second part of the flight, the differences between the channels increase significantly, and the NPF criteria indicates NPF events. At the highest altitude of 11.3 km (215 hPa) the NPF criteria (Eq. 2) was permanently fulfilled,



suggesting that the aerosols in this layer are most likely freshly formed. When calculating the particle number concentration $N_{11-15}$ in this altitude between channel 1 and channel 3 we get an average concentration of $\sim 1000$ scm$^{-3}$ for the whole flight level. Compared to the first flight level (8 km, 330 hPa) this is 4 times higher. An estimation of the particle losses in the inlet line of the Learjet and the mc-CPC for the highest flight level gives particle losses of ~22% for 12 nm aerosols (Weiden et al., 2009). The highest loss rates occur for the smallest particles (75% for 3 nm aerosols), due to the high diffusion losses at low pressures. Therefore, it is likely that the concentration of nucleation mode particles are much higher than we measured. Figure 9 demonstrates that even though the differences in the cutoffs of the individual channels are rather small, we are still able to differentiate between them and more importantly, to identify possible NPF events.

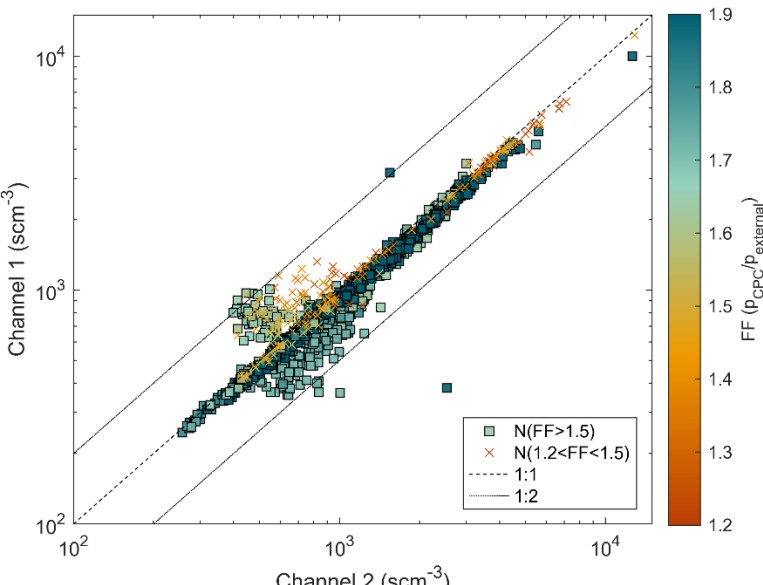

**Figure 10: Scatter plot of channel 1 and 2 (d$_{50}$ of 11.3 nm and 12.3 nm) from RF04, as 10s average data. The squares represent data points collected during the flight at FF > 1.5 and the crosses show data in a FF range of 1.2 to 1.5. Both figures are color-coded by the Flow factor (p$_{exhaust}$/p$_{CPC}$), measured by the mc-CPC peripheral.**

In Fig. 10 the aerosol concentration measured by channel 1 and 2 as 10s averages are presented as a scatter plot. All data points measured at FF > 1.2 were included in the figure. Here, we clearly see that the deviations between the channels are frequently rather high, in some cases exceeding a factor 2. Especially in a concentration regime of $300 - 1000$ scm$^{-3}$ these variations are pronounced. The color bar demonstrates that the deviations are high especially for phases were FF is small (yellow crosses). By excluding FF values that are smaller than 1.5 we gain an excellent correlation of $r^2 = 0.96$. Almost 80% of the data points at FF > 1.5 are within the uncertainty of 20%. Individual data handling is needed, when deviations exceed this uncertainty. Nevertheless, for critical flows (FF > 1.9) the data points lie almost exclusively on the 1:1 line. This finding supports our assumption that the flow fluctuations caused most of the variation in the data between channel 1 and 2. We will focus on keeping the Flow Factor > 1.9 in future aircraft campaigns to avoid these fluctuations.





**6    Conclusion**

We designed and constructed a multi-channel CPC for aircraft measurements from three individual SKY-CPCs realizing different cutoff sizes of 11.3 nm (channel 1), 12.3 nm (channel 2) and 14.9 nm (channel 3), where channels 1 and 2 have the same cutoff within their uncertainties. We chose channel 1 and 2 to be redundant in order to provide additional checks of the performance and reliability in flight. To keep the pressure in the instrument constant despite changing altitudes, we installed a
pressure regulating bypass system. We performed comprehensive mc-CPC calibrations with two independent setups at GUF and FZJ, comparing two characterization methods. Both calibrations investigated the influence of different internal CPC pressures on the counting efficiency and the cutoff diameter. The results of both calibrations showed that the cutoff is only to a small degree dependent on $p_{CPC}$ in the range of 200 – 350 hPa. Moreover, we showed that the cutoffs were comparable for both experiments, even though the setups differed in several aspects. Another observation from the GUF and FZJ calibration
is the correlation of the performance of channel 1 and 2 with the internal pressure: the higher the pressure, the higher the counting efficiency and the lower the cutoff. For channel 3 ($\Delta T$ 15°C) the trend reversed for pressures > 400 hPa. This behavior was observed for measurements with and without the pressure-reducing orifice. The use of the mc-CPC in this arrangement is therefore only suitable for pressures between 200 hPa and 400 hPa, as otherwise, the cutoffs will diverge too much and a comparison between the channels becomes difficult. This makes the pressure-reducing orifice inevitable. Still, the diverging
cutoff diameters at internal pressures > 400 hPa could be a benefit for future ground-based measurements, as changes in the cutoff diameter could be rather quickly realized by changing the internal pressure. Therefore, a comprehensive examination of the pressure-cutoff dependency at different $\Delta T$ could be of further interest. Furthermore, the influence of the pressure difference $\Delta p$ between internal and external CPC pressure on the CPC performance was investigated. The results showed that although the counting efficiency is higher when $\Delta p$ is small, the cutoffs did not change. This implies that the application of the
mc-CPC is not only possible for upper tropospheric but also for ground-based measurements (with $p_{external} \sim 1000$ hPa) and only needs minor corrections.

In order to improve the performance of the mc-CPC for upcoming campaigns, we plan a few adjustments. One thing is to shorten and straighten the individual inlet lines outside the mc-CPC housing to decrease particle line losses. To reduce the particle losses even further, the orifice changing ball valve could be replaced by a valve that switches between a pressure-
reducing orifice and an 8 mm tube. By this adaption, one can switch between a constant pressure stage and a free-floating instrument, depending on the flight level.

The mc-CPC was operational for the first time during the TPEx aircraft campaign. Research flight 04 showed a stable internal mc-CPC pressure during all flight levels. The aerosol measurement data during TPEx was affected by non-critical and thus fluctuating sample flows. For future campaigns, it is recommendable to use a separated pump for the mc-CPC to avoid flow
and pressure fluctuations. Nevertheless, after data correction and data flagging the measurements of channel 1 and 2, which were both operated at a $\Delta T$ of 36°C show an excellent agreement ($r^2 = 0.96$).



**Data availability**

Data will be put into a public Zenodo repository before final publication.

**Author contributions**

SR wrote the paper, did the data analysis and the plots. TK, LM, MH and SR designed and constructed the mc-CPC. JC conceived the project. SR performed the mc-CPC characterization setup and the corresponding measurements at GUF. PW

and SK performed the calibration setup at FZJ. PW, SK and SR carried out the measurements at FZJ. LB, MH, JS and JC gave valuable input regarding the characterization in the lab. SR, TK and JS prepared and operated the mc-CPC during the TPEx campaign. All authors commented on the paper.

**Competing interests**

One Co-author is member of the editorial board of the journal Atmospheric Measurement Techniques.

**Acknowledgements**

We acknowledge funding by the Deutsche Forschungsgemeinschaft (DFG, German Research Foundation) – TRR 301 – Project-ID 428312742: "The tropopause region a changing atmosphere", research project A03. We want to thank Peter Hoor,

Heiko Bozem and the enviscope team for the TPEx organization. A special thanks to Philipp Joppe, who operated the mc-CPC during the TPEx flight RF04 and all other contributors who participated in the campaign. We also want to thank Ralf Weigel who provided us with a test sample of FC-43. We further want to thank the ICE-3 at FZJ for the cooperation and for using the IAGOS calibration lab.






**Appendix A: Dimensions of the Learjet aerosol inlet and the mc-CPC inlet**

The Table summarizes the inlet line length inside and outside the mc-CPC housing. Combining the dimensions of all inlet lines this results in an inlet line of 143 cm for channel 1, 131.5 cm for channel 2 and 145 cm for channel 3 with an uncertainty of ± 8 cm respectively.


**Table A1: Dimensions of the individual CPC inlet lines with reading error, the common mc-CPC inlet and the Learjet aerosl inlet**.

| Inlet line | L inside housing (cm) | L outside housing (cm) | $L_{total}$ (cm) |
|---|---|---|---|
| Channel I | 28 (± 2) | 30 (± 2) | 58 (± 4) |
| Channel II | 13 (± 2) | 33.5 (± 2) | 46.5 (± 4) |
| Channel III | 42 (± 2) | 18 (± 2) | 60 (± 4) |
| mc-CPC inlet | | | 23 (± 4) |
| Learjet aerosol inlet | | | 62 |


**Appendix B: Flow correction and uncertainties**

Due to non-critical sample flows in the individual mc-CPC CPCs, we applied a flow correction to all data that were collected at FF < 1.9. This was done after the campaign. Here we measured the sample flow of each CPC with a TSI flowmeter (Series 5200) at different internal pressures and FFs. The pressure inside the CPC was changed according to $p_{CPC}$ during the TPEx camaign. The Flow Factor variied from 1.1 to > 1.9. This gives us a flow rate measured by the TSI flow meter that corresponds to a specific flow factor, which can be derived from equation 1. With this correlation we can estimate a correction factor for each CPC and pressure (see Fig. B1 b). The correction factor for three different flows are highlighted in Table B1. The pressure $p_{CPC}$ did not affect the flow rate in the CPCs when FF was constant (Fig. B1 a). Therefore all measurements for a wide range of flows and four different inlet pressures are combined to one fit function that is applied for every laboratory and campaign data set.


**Table B1: Examplary flow factors (determined by the pressure regulation system), corresponding flows (measured by TSI flow meter) and their correction factors.**

| Flow Factor ($p_{CPC}/p_{exhaust}$) | CPC flow (lpm) | Correction factor |
|---|---|---|
| > 1.9 | 0.57 (±0.04) | 1.00 (±0.01) |
| 1.8 | 0.55 (±0.04) | 1.03 (±0.02) |
| 1.5 | 0.51 (±0.04) | 1.11 (±0.06) |
| 1.2 | 0.39 (±0.03) | 1.49 (±0.26) |




In Fig. B1 a it is obvious that the sample flows measured at 200 mbar internal CPC pressure are mostly lower than for higher pressures. We cannot explain this behavior by any physical means and therefore did not account for it in the correction factor.

The correction factor $k_{FF}$ was calculated by normalizing the flow of measurement series i to the maximum flow (which should be ~ 0.6 lpm).

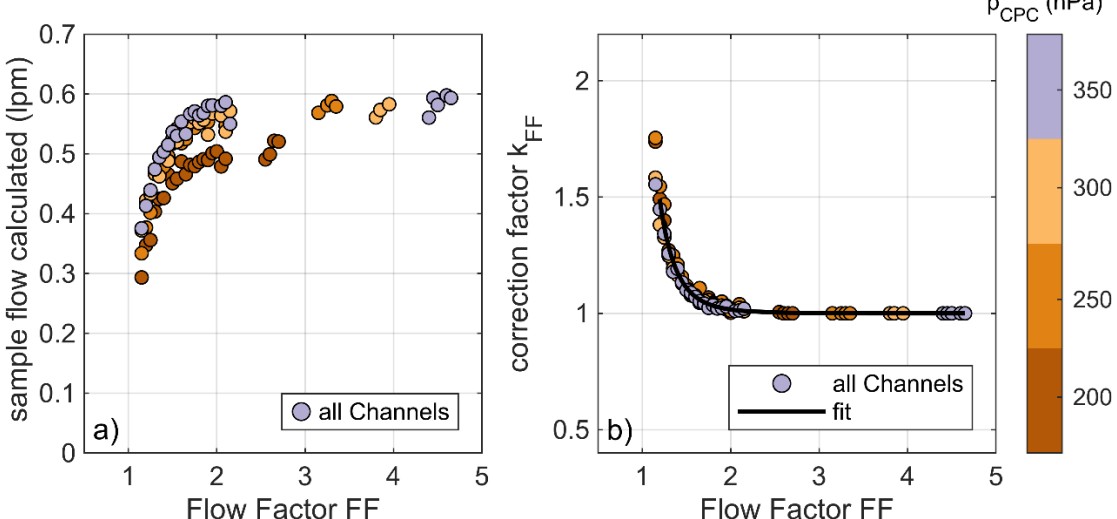

**Figure B1: a) sample flow of all three mc-CPC channels combined measured by a TSI flowmeter against the Flow**
**Factor FF, b) correction factor $k_{FF}$ (= flow$_{max}$/flow) derived from the sample flow, color coded by the internal pressure. The measurements were done at GUF. The data points represent binned data.**

The following fit function was derived from the data depicted in Figure B1 and was applied to all lab and flight measurement data collected at FF < 1.9 (excluding data points at FF < 1.2):


$$N_{i,corr} = N_i \cdot (1.6221 \cdot FF_i^{-6.6206} + 1) = N_i \cdot k_{FF} \quad with \, i = Channel \, 1,2,3 \qquad (1)$$

The uncertainty of the flow factor FF was determined to a relative error of 8%, which results out of a reading and an instrumental error of $p_{exhaust}$ and $p_{CPC}$ . With this a $\Delta k_{FF}/k_{FF}$ was estimated to a value of 19%. The overall error for the aerosol
number concentration $N_{i,corr}$ can be determined by the following:

$$\frac{\Delta N_{i,corr}}{N_{i,corr}} = \sqrt{\left(\frac{\Delta N_i}{N_i}\right)^2 + \left(\frac{\Delta STP}{STP}\right)^2 + \left(\frac{\Delta k_p}{k_p}\right)^2 + \left(\frac{\Delta k_{FF}}{k_{FF}}\right)^2} = 0.22, \qquad (2)$$



This includes an instrumental error of the aerosol concentration of 10%, an error resulting from the STP correction and an error that comes from the pressure adjustment which was determined by the plateau efficiency. The error is an upper limit, for higher *FF*s, it becomes smaller.


## Appendix C: Cutoff determination

Many research groups are using the adapted exponential fit defined by Wiedensohler *et al.* (2018) to determine the CPC specific parameters:

$$\eta_i(d) = n_{100} \cdot \left(1 - \exp\left(-\frac{d - d_0}{d_{50} - d_0} \cdot ln(2)\right)\right) \tag{3}$$

To compare whether the exponential fit or the sigmoidal fit represents the calibration data more properly we present the data here using both methods. Figure C1 shows the counting efficiency of channel 3 measured without pressure-reducing orifice.

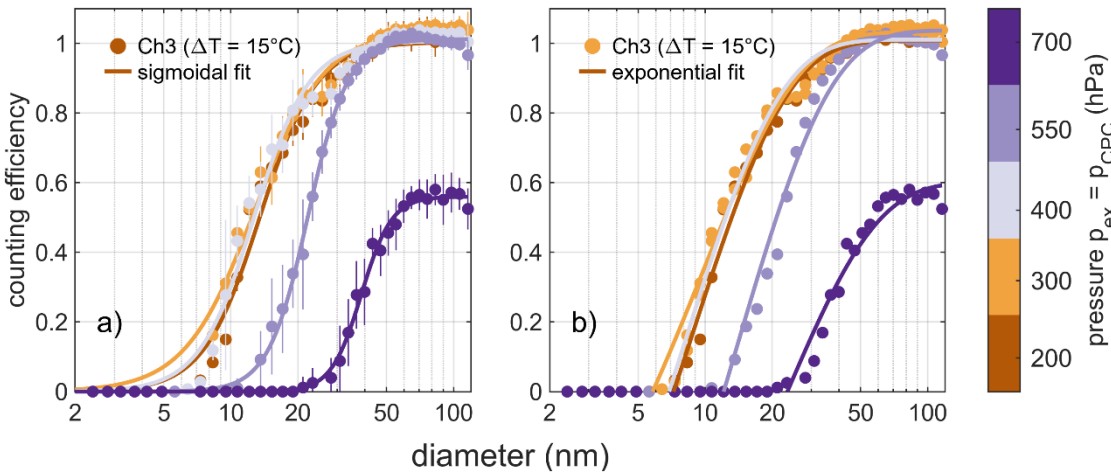

**Figure C1: Counting efficiency of channel 3 measured without mc-CPC pressure-reducing orifice at FZJ at different pressures (colorbar). In a) the data is represented through a sigmoidal fit and in b) with an exponential approach.**

Even through the cutoffs and plateau counting efficiencies derived from Fig. C1 a) and b) do not differ much, the sigmoidal
fit represents the progression of the data more accurately than the exponential fit.

## Appendix D: Cutoffs during GUF and FZJ measurements

Here the cutoff diameters and the counting efficiencies of all three mc-CPC channels and for all measurements done at GUF and FZJ are depicted. The following plots show the raw data of the counting. As a reference instrument we used the FZJ FCE.
The corresponding cutoff diameters and plateau efficiencies are listed in Table D1 and D2.





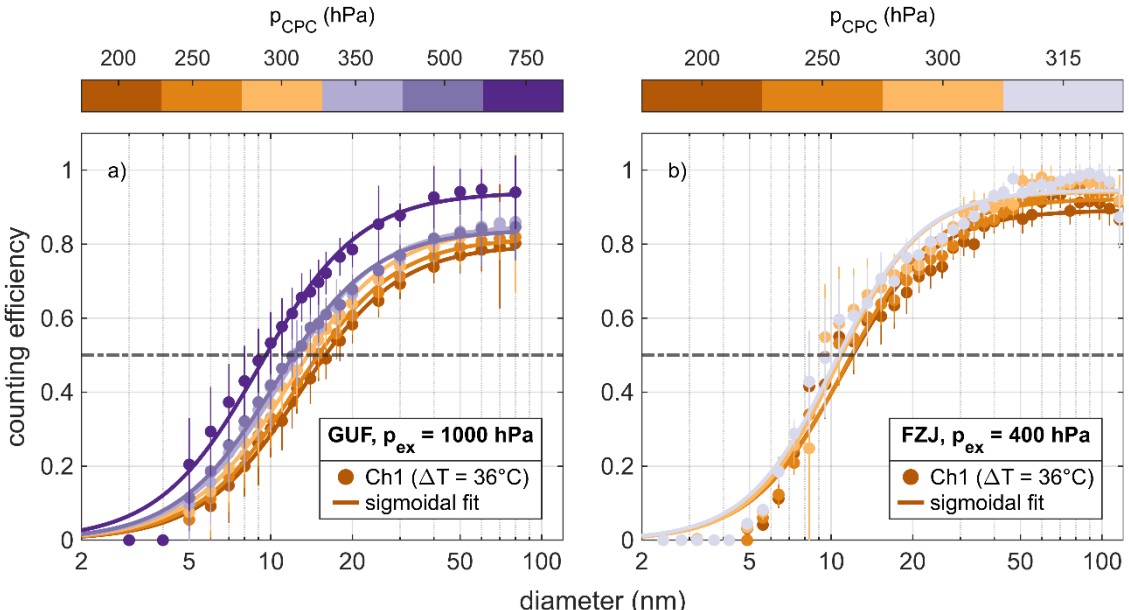

**Figure D1: Counting efficiency for channel 1 of the mc-CPC determined at a) GUF for different p$_{CPC}$ values and a fixed p$_{external}$ value of 1000 hPa and b) FZJ for different p$_{CPC}$ and a fixed p$_{external}$ of 400 mbar.**

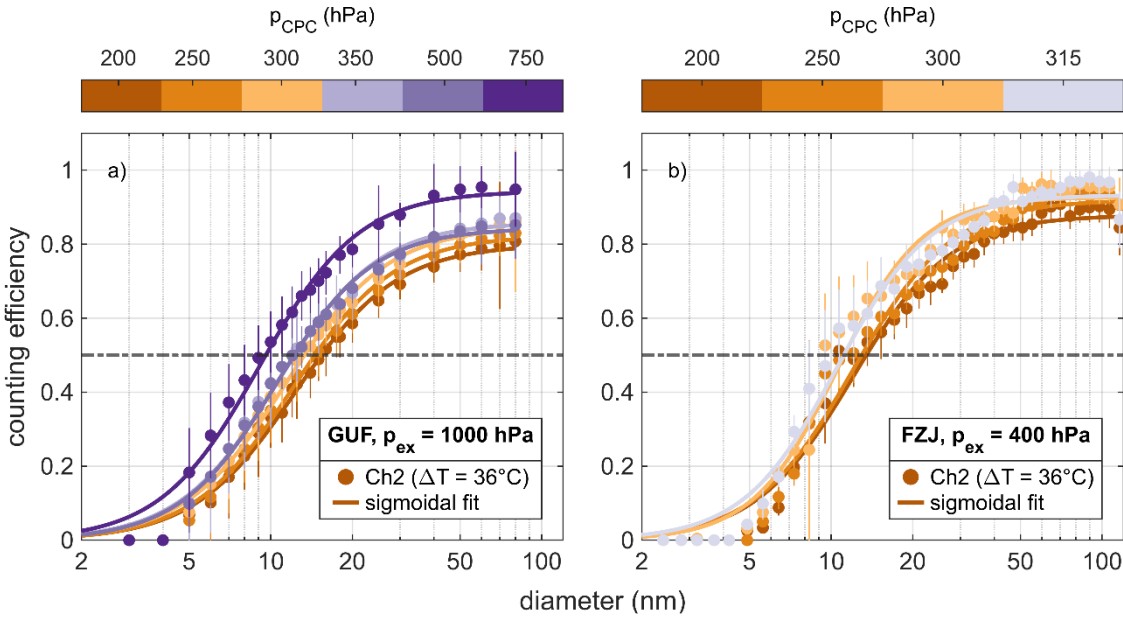


**Figure D2: Counting efficiency for channel 2 of the mc-CPC determined at a) GUF for different p$_{CPC}$ values and a fixed p$_{external}$ value of 1000 hPa and b) FZJ for different p$_{CPC}$ and a fixed p$_{external}$ of 400 mbar.**





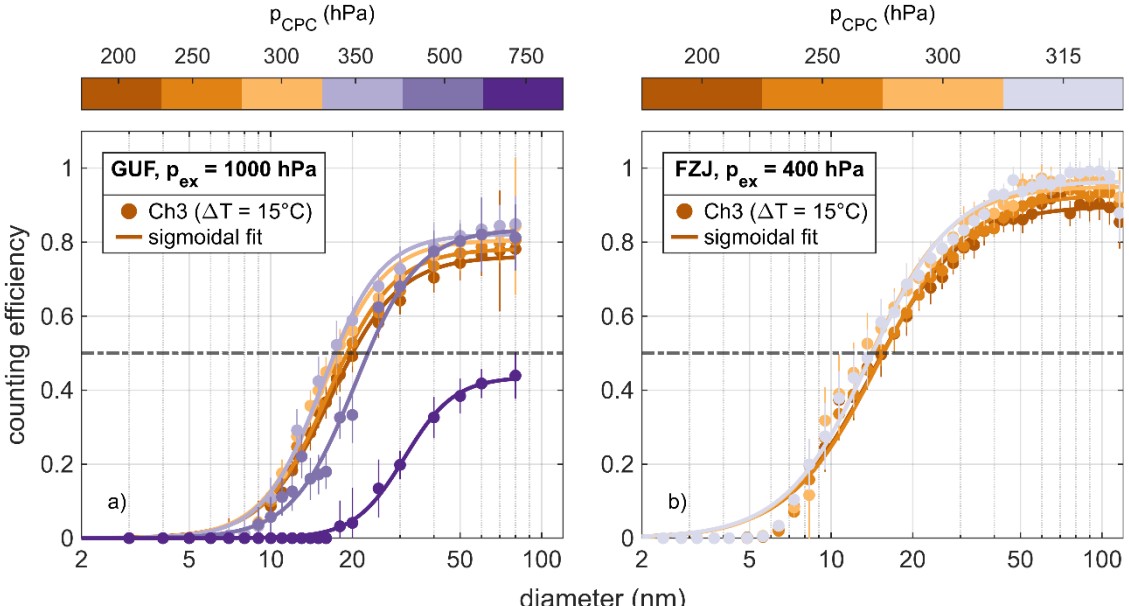


**Figure D3: Counting efficiency for channel 3 of the mc-CPC determined at a) GUF for different $p_{CPC}$ values and a fixed $p_{external}$ value of 1000 hPa and b) FZJ for different $p_{CPC}$ and a fixed $p_{external}$ of 400 mbar.**

**Table D1: cutoff diameter and plateau efficiency for all three mc-CPC channels derived at GUF for different internal**
**CPC pressures $p_{CPC}$ and at a constant external pressure $p_{external}$ of 1000 hPa.**

GUF measurements

| | Channel 1 ($\Delta T = 36°C$) | | Channel 2 ($\Delta T = 36°C$) | | Channel 3 ($\Delta T = 15°C$) | |
|---|---|---|---|---|---|---|
| $p_{cpc}$ (hPa) | $d_{50}$ (nm) | $\eta_{max}$ (%) | $d_{50}$ (nm) | $\eta_{max}$ (%) | $d_{50}$ (nm) | $\eta_{max}$ (%) |
| 200 | 12.8 (±1.3) | 80 | 12.3 (±1.3) | 80 | 16.6 (±1.8) | 78 |
| 250 | 12.4 (±1.3) | 82 | 12.1 (±1.3) | 82 | 16.4 (±1.7) | 80 |
| 300 | 11.7 (±1.2) | 85 | 11.4 (±1.2) | 85 | 15.5 (±1.6) | 83 |
| 350 | 10.9 (±1.1) | 85 | 10.6 (±1.1) | 86 | 15.4 (±1.6) | 84 |
| 500 | 10.3 (±1.1) | 85 | 10.5 (±1.1) | 83 | 20.5 (±2.2) | 85 |
| 750 | 9.1 (±1.0) | 93 | 9.1 (±0.9) | 93 | 31.2 (±3.4) | 44 |





**Table D2: cutoff diameter and plateau efficiency for all three mc-CPC channels derived at FZJ for different internal CPC pressures $p_{CPC}$ and at a constant external pressure $p_{external}$ of 400 hPa. At $p_{CPC}$ = 250 hPa two measurement cycles are averaged.**

| | FZJ measurements | | | | | |
| | Channel 1 ($\Delta T = 36°C$) | | Channel 2 ($\Delta T = 36°C$) | | Channel 3 ($\Delta T = 15°C$) | |
| $p_{cpc}$ (hPa) | $d_{50}$ (nm) | $\eta_{max}$ (%) | $d_{50}$ (nm) | $\eta_{max}$ (%) | $d_{50}$ (nm) | $\eta_{max}$ (%) |
|---|---|---|---|---|---|---|
| 200 | 10.9 (±0.9) | 89 | 11.9 (±1.0) | 88 | 14.2 (±1.2) | 90 |
| 250 | 11.2 (±1.0) | 92 | 12.0 (±1.0) | 92 | 14.7 (±1.3) | 93 |
| 300 | 10.5 (±0.9) | 94 | 10.6 (±0.9) | 93 | 13.6 (±1.2) | 95 |
| 315 | 10.2 (±0.9) | 95 | 10.5 (±0.9) | 93 | 13.9 (±1.2) | 97 |


## Appendix E: Counting efficiencies at different ambient pressures

The raw data of all three mc-CPC channels measured at a fixed $p_{CPC}$ and a variable $p_{external}$ are depicted in Fig. E1. Here we can see that the asymptotic counting efficiency is lowered when the difference between ambient and CPC pressure is high.

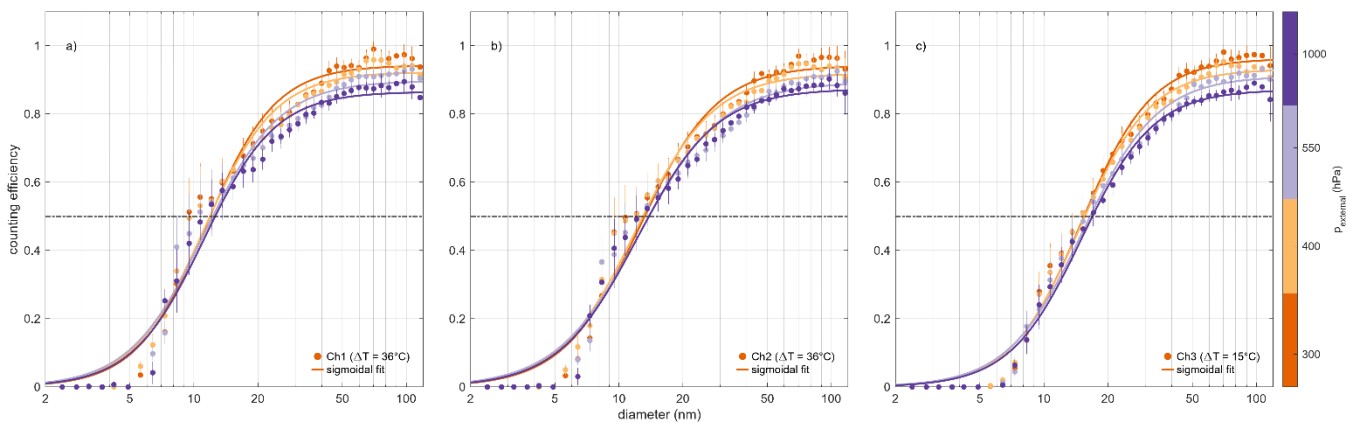


**Figure E1: Raw counting efficiencies of all three mc-CPC channels (a – c), colored by the external pressures. The internal pressure $p_{CPC}$ is 250 hPa for all panels. The error bars represent the standard deviation. Note that for $p_{external}$ = 300 and 400 hPa the depicted data is averaged for two measurememnt cycles.**




**Appendix F: Counting efficiency for FZJ**

In Fig. F1 all mc-CPC calibrations and the determined counting efficiency at FZJ are illustrated. The Figure shows that the
higher the difference between the pressures inside and outside of the mc-CPC, the lower the counting efficiency.

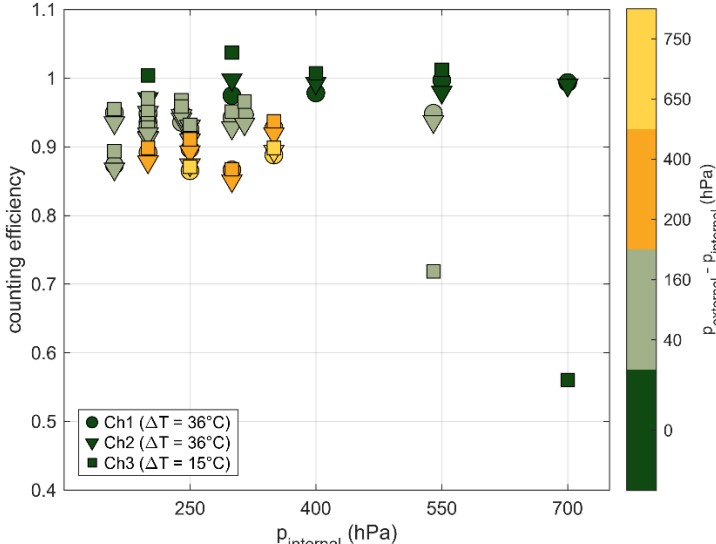

**Figure F1:  Counting efficiency for all three mc-CPC channels at various internal CPC pressures, colored by four distinct Δp.**




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
