# Peer review of "Characterization and operation of a multi-channel Condensation Particle Counter (mc-CPC) for aircraft-based measurements"

_EGUsphere, 2025_

## Author Comment (AC1)

**AMT manuscript egusphere-2025-4349**

Characterization and operation of a multi-channel Condensation Particle Counter (mc-CPC) for aircraft-based measurements, S. Richter et. al

**Author comments to reviewer #1**

*The comments of the reviewer are depicted in black and italics.*

Our answers to the reviewer comments are written in green color.

Changes in the revised version of the manuscript are given in red color.

**Referee comment #1**

*This paper presents a description of an aircraft-based, multi-channel condensation particle counter (mc-CPC) used for investigating the new particle formation (NPF) in the UTLS. They provide a detailed description of the system design and the careful, comprehensive calibrations, with an example of in-flight data from the TPEx campaign on tropopause composition in 2024. This mc-CPC system was developed by integrating three commercial GRIMM SKY-CPCs with a custom-built pressure regulation and flow manifold for aircraft-based measurements. FC-43 was used as the working fluid, which, according to the authors, was tested for the first time on Grimm SKY-CPC. The mc-CPC system consists of three channels to provide two size cuts, ~11-12 nm (chan1 and chan2) and ~15 nm (chan3). The counting efficiency of the CPC was corrected for flow and pressure, but not for the particle loss through the inlet and sampling line. The comparative use of the two size cuts from the mc-CPC (i.e., the difference between the particle number concentrations of the low and high size cuts) provides identification of NPF events.*

We thank the reviewer for the detailed and comprehensive comments, which are very valuable and will improve the manuscript.

*Main comments:*

1. *Wording of "construct". "We constructed a multi-channel condensation particle counter." I am not sure if the wording of "construct" is entirely accurate here. It seems more like a custom integration built around commercial CPC units.*

We agree that the word 'construct' is not really accurate. We therefore changed the manuscript accordingly. The lines listed below refer to the revised version.

Line 12 & Line 702: changed 'constructed' to 'set up'.

Line 76-79: Using a similar approach, we set up a custom integration of three commercial CPC units for aircraft applications. The three channels of the multi-channel Condensation Particle Counter (mc-CPC) are currently operated with FC-43 (Fluorinert) as the working fluid and provide two different cutoffs by adjusting the internal CPC temperatures.

Line 82: changed 'construction' to 'design'.

Line 87: Deleted 'construction of the'.

Line 13, 87 and 141: Added 'commercial' in front of 'CPC'.

> 2. *The biased particle concentration is a concern. The absolute particle number concentration from mc-CPC cannot reflect the true ambient values because the particle loss in the inlet and tubing is not corrected. The authors have attempted to estimate the particle loss during NPF events, but because the particle size distribution is unknown, the actual concentrations are still quite uncertain. The particle concentration of the two size cuts can be used comparatively for NPF event identification, because all CPCs share the same common inlet and sampling line. However, the loss of particles is heavily dependent on their size, especially for particles smaller than 20nm; as a result, the errors in absolute concentration between the two size cuts could bias the identification of NPF events.*

We agree that a proper correction to derive the ambient number concentration is not possible because the size distributions are not known. We are nevertheless convinced that the particle losses do not strongly affect the validity of the NPF identification. First, the particle losses for the small and the large channel are comparable because they share the same inlet (26% for 11 nm and 20% for 15 nm-sized particles). Furthermore, the diffusion losses for particles of 11 nm are higher than for the 15 nm particles. This means, that by not accounting for the differences in size-dependent inlet line losses we rather underestimate the differences between the two channels, which means that the NPF criteria we use is even more conservative. To make this clearer we added the following paragraph in section 5, line 667:

As we could not perform a quantitative particle loss correction because of the unknown size distribution, the measured concentrations represent lower limits of the ambient aerosol concentration. Nevertheless, as all channels are subject to similar particle losses due to their common inlet, the identification of NPF events should not be affected strongly. Furthermore, the general concentration range and relative trends of the total concentration are well represented by the measurements.

> 3. *The size cut for "recent NPF events." The motivation of the mc-CPC is to investigate the NPF events in the UTLS, which, in this study, are identified by the difference in particle number concentrations between the lower size cut and higher size cut (12-15 nm). Can 12-15 nm (instead of sub-10 nm) be used to identify "recent NPF" (line 240)? Or maybe this is just another wording issue with the word "recent".*

We agree, freshly formed particles have to grow to sizes that are detectable by our instrument (~11 nm). The time it takes to grow to a detectable size depends on the average growth rate. These growth rates can be quite variable. For growth rates of ~ 9 nm/h as observed during strong upper tropospheric NPF events by Curtius et al. (2024), we would be able to detect an NPF event that happened a bit more than ~ 1 hour ago. The average growth rate after NPF depends on various atmospheric conditions and substantially lower growth rates are conceivable as well (e.g. Kupc et al., 2020). However, we believe that in the time frame of a few hours the word 'recent' is appropriate and we would therefore prefer not change the wording in the revised version. We nevertheless adjusted the paragraph to make this clearer (line 247):

 (…) which most likely have formed by recent NPF a few hours ago. Note that growth rates in the UT are highly variable and therefore the time between fresh nucleation and our measurements can differ.

4. *The normalized counting efficiency and data correction. The normalized counting efficiency was shown in the manuscript to illustrate the instrument/channel comparison. However, the manuscript didn't explicitly state whether the raw/absolute counting efficiency was used for in-flight data correction. There needs to be some statement to clarify this.*

We suppose you mean the characterization that we performed for different internal and external pressures and the resulting change in the counting efficiency (4.4). Yes, the in-flight data has been corrected for this offset. Therefore, we added a paragraph at the end of this section (line 575 in the revised version).

For our aircraft measurements, this means that varying altitudes do not alter the cutoff but the plateau efficiency changes slightly with altitude. Hence, the measurement data of the research flight presented in section 5 were corrected by the raw counting efficiency according to the $p_{CPC}$ and the ambient pressure $p_{external}$ (see Fig. G1 for the summarized data).

5. *The Correction factor at 200 hpa (Pcpc). The internal pressure of the CPC (Ppcp) ranges from 200 hpa up to 750 hpa. However, the sample flow at 200 hpa is lower than that at higher pressures, particularly for FF>1.5 (line 743-744, Fig. B1). In the manuscript, the author stated that they cannot explain this behavior and did not account for these lower flows in the correction factor. However, the mc-CPC was operated at ~ 200 hpa for most of the time (i.e., RF04). If 200 hpa is a typical operating pressure for mc-CPC, and the outstanding flow behavior at 200 hpa is consistent, it needs further investigation and should not be simply ignored.*

We thank the reviewer for this comment. We agree that the lower sample flows observed at $p_{CPC}$ 200 hPa needs further investigation. We would like to emphasize that we did not intend to ignore the observed behavior and that we pointed it out in line 741 (line 815 in the revised version). We repeated measurements at $p_{CPC}$ = 200 hPa several times at different days and the behavior was unfortunately inconclusive: sometimes the correct flow of 0.6 lpm was observed (fitting perfectly to the flows at higher pressures) and sometimes lower flow rates occurred. The flows shown in Fig. C1 show the average of these measurements. We do not see a reason for the changing flows and are convinced that this is most likely an artifact of the flow measurements. To make this clearer we added a paragraph to Appendix C.

Line 816: The flows for $p_{CPC}$ = 200 hPa as depicted in Fig C1 are lower than the 0.6 lpm that were observed for higher values of $p_{CPC}$. The measurements shown represent an average of several measurements, and for some measurements also a flow of 0.6 lpm was observed, as expected. We think that the sometimes lower flows are actually an artefact, but we were not able to fully resolve this issue with the available instrumentation. Still, this issue needs further investigation in the future.

6. *The manuscript could benefit from being more focused and concise, emphasizing the key points and minimizing redundancy.*

We agree that the manuscript is fairly long and has some redundancies. To address the reviewers comment we deleted or rephrased text:

Deleted text:

Line 196: The inlet flow (…)

Line 317: The comparison of these two parameters (…)

Line 436: To examine (…)

Line 483: We will use cutoff diameters (…)

Line 586: An aerosol number concentration correction (…) → added text to 577

Line 604: These values represent mean values (…) → added to Fig. 8 description

For more, please see the revised manuscript.

Rephrased text:

Line 192: The high flow rates that the IDP-3 pump had to provide during TPEx caused some difficulties (…)

Line 503: By increasing $p_{CPC}$ to 700 hPa we possibly also decreased the diffusion of FC-43 into the center of the saturator, which consequently could have had an unfavorable effect on the particle activation.

Line 564: (…) while the deviations for channel 2 and 3 were somewhat larger, ranging from 12 nm to 12.6 nm and from 14.7 nm to 15.2 nm, respectively, which is still within their uncertainty.

For changes in section 4.2 and 4.3 please see the revised version.

***Other comments***

145    *1. Need to keep consistency for the use of terms "NMP", "NMPs", and "nucleation mode particles". For example, line 56 uses both NMP and nucleation mode particles.*

For the revised manuscript we changed the abbreviation NMP into NMPs, because we only refer to them in the plural form.

Line 56: Various sources of NMPs in the UT exist, but they are dominated by local production
150    (…).

   *2. Line 79: a constant low pressure? According to the manuscript, the cpc pressure was regulated but not constant. Or does it set at a constant pressure for each flight? Please be clearer here.*

Many thanks, this was indeed not formulated concisely. We changed the respective lines:

155    A pressure regulation system with a critical orifice ensures a low pressure in the system. The set point was adjusted according to the flight pattern and therefore varied between 200 hPa and 350 hPa.

   *3. Table 1: might need to list the constants used in the Antoine equation for Butanol and FC-43, and the corresponding references.*

160    The Antoine equations for Butanol and FC-43 as well as the used constant are now included into the Appendix A of the revised manuscript.

**Appendix A: Antoine equation**

To calculate the vapor pressure $p_{vap}$ of butanol in the CPC, we used the following equation with the corresponding parameters b = 46.78 and c = 11.26 (Baron & Willeke, 2001), where T
165    is given in Kelvin and can be replaced by the CPC temperatures $T_{sat}$ and $T_{con}$.

$$log_{10}(p_{vap}) = \frac{-52.3 \cdot b}{T} + c \qquad (1)$$

For the vapor pressure of FC-43 dependent on the saturator and condenser temperature, the following equation was used (Baron & Willeke, 2001; 3M, 2019):

$$log_{10}(p_{vap}) = a - \frac{b}{T} \qquad (2)$$

170    Here the parameters a and b are determined to 10.511 and 2453, respectively (3M, 2019).

   *4. Line 164: "a low and constant pressure", was the pressure here meant for P1 or P2? Need to clarify here.*

Line 170: (…) to maintain a low and constant pressure at P1.

175    *5. Figure 1: Incomplete information, for example, the label of the IDP-3 pump is missing.*

Thanks for the thorough examination of the figures! In this case we added the label "IDP-3" to the pump and also changed the description "inlet" into "Common inlet".

*6. Mentioned the full name of the TPEx campaign multiple times in the manuscript (i.e., line 26, line 80, and line 170); only needs to mention the full name once to reduce*
180   *redundancy.*

Thanks. We kept the full name in the abstract (line 26) and removed it in line 81 and 176. We also deleted "TPEx campaign" several times in the script (e.g. line 144, line 317 etc.).

*7. Figure 2: For the cold reservoir of alpha pinene–shouldn't the left tube insert deeper than the right tube?*

185   Actually, there are no tubes at all that reach fully into the alpha pinene reservoir. Because this was not correctly represented in Figure 2, we changed the schematic accordingly and removed the tubes.

*8. Line 587-589: "We tentatively propose that altering diffusion rates in combination with the long mc-CPC inlet lines could have caused the dropping CPC performance with*
190   *decreasing pressures." Was it "decreasing pressure" or "increasing pressure"? Was the "dropping CPC performance" meant the deviated size cuts at increasing pressure for channel 3 (Fig. 8a)?*

With this we were referring to the pressure range between 200 and 400 hPa where all three channels show a similar behavior; lower counting efficiencies and larger cutoff diameters with
195   decreasing pressures. Still, as this explanation could be also the reason for the dropping channel 3 performance at higher pressures, we added the following:

We tentatively propose that altering diffusion rates in combination with the relatively long mc-CPC inlet lines could have caused the dropping CPC performance for all three channels with decreasing pressures in the range of 200 hPa to 400 hPa. However, also the increasing cutoff
200   sizes observed for channel 3 at $p_{CPC} > 400$ hPa could be a result of varying diffusion rates.

*9. Line 730: a typo here: variied → varied*

We changed "variied" to "varied".

*10. Line 744: The CPC flow is lower at 200 hpa, which is a typical operating pressure of the mc-CPC. However, here the outstanding behavior at 200 hpa was simply ignored.*
205   *Has the sample flow at this pressure been measured more than once? Is this a consistent behavior? If so, I don't think this behavior can just be dismissed from the calculation of the correction factor.*

Please see our answer to comment #5 in the "main comments" section.

210

**References**

3M Safety data sheet: 3M™ Fluorinert™ Electronic Liquid FC-43. https://multimedia.3m.com [Accessed 11 July 2025], 2019.

Baron, P.A. & Willeke, K. Aerosol measurement: *principles, techniques, and applications*, 20th edition. Wiley-Interscience: New York, XXIII, 1131 pp, 2001.

Curtius, J., Heinritzi, M., Beck, L.J., Pöhlker, M.L., Tripathi, N. & Krumm, B.E. et al. Isoprene nitrates drive new particle formation in Amazon's upper troposphere. Nature, 636(8041), 124–130. https://doi.org/10.1038/s41586-024-08192-4, 2024.

Kupc, A., Williamson, C.J., Hodshire, A.L., Kazil, J., Ray, E. & Bui, T.P. et al. The potential role of organics in new particle formation and initial growth in the remote tropical upper troposphere. Atmospheric Chemistry and Physics, 20(23), 15037–15060. https://doi.org/10.5194/acp-20-15037-2020, 2020.

---

## Author Comment (AC2)

**AMT manuscript egusphere-2025-4349**

Characterization and operation of a multi-channel Condensation Particle Counter (mc-CPC) for aircraft-based measurements, S. Richter et. al

**Author comments to reviewer #2**

5 *The comments of the reviewer are depicted in black and italics.*

Our answers to the reviewer comments are written in green color.

Changes in the revised version of the manuscript are given in red color.

10 **Referee comment #2**

We thank the reviewer a lot for the thorough reading and the valuable comments and questions, which will help to improve this manuscript. We especially appreciate the proposed changes to the mc-CPC design that will increase the performance.

*Excellent work in characterizing and correcting the performance of the CPC during non-ideal*
15 *cases, such as unknown/uncontrolled sample flow and varying external and instrument pressures. There's a lot of normalization that is happening that probably warrants a summation of error bars to understand the compounding corrections applied to the field dataset.*

The summation of the error bars including the uncertainties of the normalization factors is given in Appendix C.

20 *The equation determining NPF is maybe overly strict, but also seems arbitrary as it's currently written. Considering this is the major focus of the instrument, I think this needs better justification or characterization. I think the combination of the aforementioned error analyses and Poisson variance would provide a stronger basis for quantifying NPF.*

See below for our reply, why we think that using our fairly conservative NPF criteria is
25 preferable when taking also systematic uncertainties into account. We are trying to reduce these uncertainties for future aircraft measurements, and then we will consider to switch to the Poisson statistics criteria for determining the presence of NPF events.

*Without the inlet pressure control being appropriately designed and characterized, the instrument may not accurately indicate the total particle number concentration, but is sufficient*
30 *for NPF quantification since common-mode errors of larger particle loss is subtracted. This can be easily remedied in the next instrument update, to improve the amount of information received. Similarly, the second channel can be put to better use gathering additional information instead of redundant with the first channel.*

We fully agree. We are eager to a) reduce the inlet losses by using a properly designed pressure
35 control unit, b) obtain more information on the aerosol size distribution (from other instruments running in parallel) and therefore being able to determine the size-dependent losses, and c) changing the cut-off settings for the redundant channel. The redundancy for channels 1 and 2 was only intended for the initial operation to gain confidence about the reliability of the absolute measurements.

40    *L020: Why was flight 4 selected when it was operated at a different pressure from typical?*

We decided to show this flight in particular because we think that it is most suitable as a proof of concept. Our main goal was to present a flight that includes a distinct NPF event, which was only observed in a subset of flights. Furthermore, we wanted to show a flight that also includes non-ideal measurement conditions (flow fluctuations) to gain a better understanding of the data
45    quality. Here, Flight 4 was the best choice although the pressure setting was indeed not the typical 250 hPa.

*L089: Suggest changing "gets mixed" to "mixes" for succinctness.*

Thanks, we changed it to "mixes" as suggested.

*L090: Suggest changing "enable" to "causing" or "activating", showing causality.*

50    We changed "enable" to "activating".

*L101: Critical orifice is not shown in Figure 1 flow diagram. Is it within the Channel block? What is the role of the pump valve if not for flow control?*

Yes, the critical orifice that is mentioned in this paragraph is indeed located inside the channel block. The pump valve represents a safety measure and can be either open or close. As it is
55    located outside the channel block it is also shown like this in Figure 1. We added the following to line 106 of the revised manuscript:

(…) which is located downstream of the detection cell inside the channel block.

*L221: Correcting for inlet orifice particle loss isn't necessary if only subtracting channels for an NPF measurement, but it is critical if trying to observe total particle concentration,*
60    *especially in the lower Stratosphere, where mode size is typically accumulation mode. During ACCLIP, the integrated UHSAS concentration often exceeded the NMASS concentration in the Stratosphere due to losses in the pressure control orifice. Losses can be reduced with an appropriately designed expansion section after the orifice.*

We agree that the absolute ambient particle number concentrations cannot be derived from our
65    measurements, only lower limits. This is something we plan to improve for future flights and therefore we are grateful for your suggestions about including an appropriately designed expansion section after the orifice. To make this clearer we added the following to section 5, line 667 of the revised version:

As we could not perform a quantitative particle loss correction because of the unknown size
70    distribution, the measured concentrations represent lower limits of the ambient aerosol concentration. Nevertheless, as all channels are subject to similar particle losses due to their common inlet, the identification of NPF events should not be affected strongly. Furthermore, the general concentration range and relative trends of the total concentration are well represented by the measurements.

75    *L241: Eqn. 2 values seem arbitrary. Some of the cited papers use different values for uncertainty. Recommend testing analysis from Williamson et al., 2019 with your dataset based on Poisson counting statistics.*

Thanks a lot for the suggestion. We reproduced the formula you suggested and determined the NPF criteria for our data with the Poisson counting statistic. When comparing channel 1 and 3 we typically get $\sigma_{diff}$ in the range of 100-200 #/cm$^3$. For our data this means that a large part of the research flight would have been classified as an NPF event, although the difference between both channels could have been caused by systematic flow differences. We believe that the Poisson approach is feasible for very robust data sets that are hardly affected by systematic errors. For the mc-CPC this is unfortunately currently not the case as the data quality was influenced by the flow fluctuations in our system. Even the channels with the same $\Delta T$ are subject to some deviations (see Fig. 10). The systematic variations resulting from instrument uncertainties could falsely indicate NPF events. Therefore we would rather keep the more conservative approach. We decided to use the 30% as an NPF threshold because the uncertainty of the individual CPCs was determined to a maximum value of 22% (Line 241). By setting the NPF criteria to 30% we wanted to avoid misclassification. If we can be confident that systematic errors like a drift of the flows is excluded, then we will change to the Poisson approach in the future.

*L328: What was the DMA sample flow? Relevant to understanding transfer function width (and thus horizontal error bars in the sizing).*

The DMA sample flow was 0.7 lpm and this was added to line 339.

Here we need to clarify that the DMA bandwidth error at FZJ was determined with a sample flow of 1 lpm. We changed this accordingly in the revised manuscript.

*Fig 4: Can you clarify how the error bars are calculated? Is it the standard deviation from the varying particle number concentration measured during the time period of each test point? Is the standard deviation of the TSI reference CPC considered, since it will fluctuate as well. In Poisson statistics, need to sum variances together.*

We calculated the efficiency error by using the standard deviation of the respective mc-CPC channel and the TSI reference instrument and combining these two errors by error propagation.

$$\frac{\Delta(eff)}{eff} = \sqrt{\left(\frac{\sigma_{mcCPC}}{N_{mcCPC}}\right)^2 + \left(\frac{\sigma_{TSI}}{N_{TSI}}\right)^2}$$

After checking all figures and the error bars once more, we noticed a minor inaccuracy of the depicted figure and in the text. We thus have updated Fig. 4 and also corrected the error bars using the above mentioned formula. We also added a part in the description of Fig. 4:

The error bars in a) represent the combined uncertainty of the counting efficiency, derived from the standard deviations of the two aerosol instruments.

*L472: This would not be a factor for sub-100 nm particles if an expansion section was included.*

Thanks a lot for this suggestion. The mc-CPC could really benefit from an expansion section after the pressure-reducing orifice. We are planning to realize such an improvement in the future.

*Fig 7: I think it would be nice to see the effect that pressure has on the plateau efficiency, rather than the normalized plots.*

This is a very valid point, as from Fig. 7 one cannot directly see the decrease in counting efficiency with increasing dp ($p_{external}$ - $p_{CPC}$). Still, in Table 4 these changes are summarized and the raw data can also be found in Appendix F Figure F1 in the revised version. As we decided to only show the normalized data for consistency, we would rather keep this form of presentation.

*Table 4: Is Ch3 operating temperature a typo, or did you operate Ch3 at dT=36C? If so, why did you calibrate the counting efficiency outside of its operating spec? And why is D50 of CPC3 relatively high for the same dT?*

This is indeed a typo. Ch3 was operated at a $\Delta T$ of 15°C throughout all measurements. Therefore we changed it in the table from "$\Delta T$ = 36°C" to "$\Delta T$ = 15°C".

*Fig 9: How can you have N11-N15 ~= 1000 /scm3 and consider it "no NPF"? This circles back to my comments on Eqn. 2. The whole section from 12:30-13:45 looks strongly related to NPF.*

Yes, we also consider that the deviation between N11-N15 of 1000 scm$^{-3}$ arises likely from NPF. This was stated in the paper as follows: "However, in the second part of the flight, the differences between the channels increase significantly, and the NPF criteria indicates NPF events". But we agree, that it is better to indicate the exact time, when we assume NPF to occur. Therefore we have changed the sentence and also added time stamps for the short NPF events:

Line 673: (…) was only fulfilled for a few seconds (e.g. 11:33).

Line 675: However, in the second part of the flight and especially in the time from 12:30 to 13:45, the differences between the channels increase significantly, and the NPF criteria indicates NPF events.

The NPF criteria shown in Fig. 9c was updated as we had used channel 2 and not channel 1 as stated in the description. We updated the plot accordingly and implemented the new figure into the revised manuscript.

*L627: Is the inlet pressure reduction orifice properly sized for 200 hPa operation? Maybe you can go smaller and switch at a higher altitude to reduce amount of bypass flow the pump has to accommodate.*

This is a very good suggestion that we will consider it for future campaigns. For this instrument we had a switchable orifice, but the second one was even larger than the one we used.

*L641: The diffusion losses of 10-13 nm can be corrected since you have that information from subtracting the two channels and their relative contribution to the total concentration.*

When we assume that the average size of the $N_{11-15}$ channel is 13 nm, then the diffusion losses can be calculated to 22%. We state this loss in the caption of Fig. 9 now, but we did not apply this as a correction factor in order to keep the $N_{11-15}$ data directly comparable with the $N_{11}$ and $N_{15}$ data.

*Instrument suggestions for future development: Designing and characterizing an appropriate expansion chamber at the inlet pressure control orifice will allow the first channel to (closer) measure a total number concentration. Now that the first and second channel have been tested*

155 *together, I think the first channel alone can be trusted, so the second channel can be used for additional information such as a third cut size or a non-volatile measurement (if inlet orifice is characterized through Accumulation mode).*

Thanks again for the very helpful remarks. We greatly appreciate your comments and suggestions for the instrument improvement. For the next adjustment of the mc-CPC we will
160 consider all these ideas.

We added a sentence to the conclusion:

Line 727: Another aim is to use all three channels at different cutoffs to gain more information about the air masses and potential NPF events.